# *Grasp55*−/− mice display impaired fat absorption and resistance to high-fat diet-induced obesity

Jiyoon Kim [1,2], Hyeyon Kim[3], Shin Hye Noh[1], Dong Geon Jang[1], Shi-Young Park [4], Dongkook Min [5], Hyunki Kim[6], Hee-Seok Kweon[7], Hoguen Kim [8], Sowon Aum[1], Sookyung Seo[1], Cheol Soo Choi[4], Hail Kim [6], Jae Woo Kim [5], Seok Jun Moon [3], Heon Yung Gee [1] & Min Goo Lee [1]✉

The Golgi apparatus plays a central role in the intracellular transport of macromolecules. However, molecular mechanisms of Golgi-mediated lipid transport remain poorly understood. Here, we show that genetic inactivation of the Golgi-resident protein GRASP55 in mice reduces whole-body fat mass via impaired intestinal fat absorption and evokes resistance to high-fat diet induced body weight gain. Mechanistic analyses reveal that GRASP55 participates in the Golgi-mediated lipid droplet (LD) targeting of some LD-associated lipases, such as ATGL and MGL, which is required for sustained lipid supply for chylomicron assembly and secretion. Consequently, GRASP55 deficiency leads to reduced chylomicron secretion and abnormally large LD formation in intestinal epithelial cells upon exogenous lipid challenge. Notably, deletion of *dGrasp* in *Drosophila* causes similar defects of lipid accumulation in the midgut. These results highlight the importance of the Golgi complex in cellular lipid regulation, which is evolutionary conserved, and uncover potential therapeutic targets for obesity-associated diseases.

[1] Department of Pharmacology, Brain Korea 21 Project for Medical Sciences, Severance Biomedical Science Institute, Yonsei University College of Medicine, Seoul 03722, Korea. [2] Department of Pharmacology, Department of Biomedicine & Health Sciences, College of Medicine, The Catholic University of Korea, Seoul 06591, Korea. [3] Department of Oral Biology, BK21 PLUS, Yonsei University College of Dentistry, Yonsei-ro 50-1, Seodaemun-gu, Seoul 03722, Korea. [4] Korea Mouse Metabolic Phenotyping Center, Lee Gil Ya Cancer and Diabetes Institute, and Department of Internal Medicine, Gachon University College of Medicine, Incheon 21999, Korea. [5] Department of Biochemistry and Molecular Biology, Severance Biomedical Science Institute, Yonsei University College of Medicine, Seoul 03722, Korea. [6] Graduate School of Medical Science and Engineering, Korea Advanced Institute of Science and Technology, Daejeon 34141, Korea. [7] Center for Research Equipment, Korea Basic Science Institute, Cheongju 28119, Korea. [8] Department of Pathology, Brain Korea 21 Project for Medical Sciences, Yonsei University College of Medicine, Seoul 03722, Korea. ✉email: mlee@yuhs.ac

Many transport vesicles containing diverse proteins and lipids travel through the endoplasmic reticulum (ER)-to-Golgi trafficking pathway. The cargo proteins and lipids then subsequently move to their final destinations, including the plasma membrane. Since the discovery of the Golgi body over 120 years ago, its function has been studied mostly from a protein-centric viewpoint[1,2]. However, a growing body of evidence suggests that the Golgi is also indispensable for regulation of cellular lipids and, in particular, plays an important role in the cellular control of lipid storage and transport[3,4]. It is well known that apolipoprotein B (APOB)-containing lipoproteins, such as chylomicrons in the intestinal epithelium and very-low-density lipoproteins (VLDLs) in hepatocytes, exit the cell via the classical ER-to-Golgi secretory vesicular pathways[5,6]. Furthermore, knowledge accumulated over the last two decades has identified some molecular machineries that are involved in the trafficking and lipidation of APOB-containing lipoproteins during their passage through the Golgi[7,8]. For example, microsomal triglyceride transfer protein (MTP) has been shown to localize to the Golgi and contribute to the maturation of APOB-containing lipoproteins[9]. However, most mechanisms underlying Golgi-mediated lipid transport pathways are not well defined.

The Golgi reassembly-stacking proteins (GRASPs/GORASPs) were initially identified in vitro as factors required for the stacking of Golgi cisternae[10,11]. However, later studies indicated that GRASPs are largely dispensable for Golgi-mediated conventional anterograde protein transport[12]. Instead, the extra-Golgi GRASPs under cellular stress conditions have been implicated in unconventional protein secretion pathways of diverse cargos that bypass the Golgi[13,14]. Two isoforms of GRASP have been identified in vertebrates, GRASP65/GORASP1 and GRASP55/GORASP2, and a recent report suggested that ablation of Grasp65 in mice did not result in phenotypic defects[15]. In addition, the phenotype of Grasp55 knockout mice has not been studied in detail, with the exception of spermatogenesis defects and male infertility[16]. Therefore, the physiological roles of GRASPs remain largely elusive.

Here, through an integrated analysis in mouse, fly, and in vitro systems, we demonstrate that the Golgi-resident protein GRASP55 plays a crucial role in lipid homeostasis. We find that genetic inactivation of GRASP55 in mice reduces whole-body fat mass via impaired intestinal fat absorption. In the mouse intestine, GRASP55 deficiency leads to reduced chylomicron secretion and abnormally large lipid droplet (LD) formation in response to exogenous lipid challenge, which is associated with failure of Golgi-mediated LD targeting of some lipases, such as adipose triglyceride lipase (ATGL) and monoglyceride lipase (MGL). The lipid absorption and accumulation defects induced by GRASP55 deficiency are rescued by the supplementation of GRASP55 in mice. Moreover, loss of dGRASP, the single Drosophila GRASP homolog, causes similar defects of lipid accumulation in the Drosophila midgut and these defects are rescued by the supplementation of dGRASP or BMM, a Drosophila homolog of mammalian ATGL[17]. Our data indicate that GRASP55 plays an essential role in intestinal fat absorption in live animals. In addition, these results suggest that GRASP55 is critical for Golgi-mediated LD targeting of key LD regulating proteins and highlight the importance of the Golgi complex in cellular lipid regulation.

## Results

### Grasp55$^{-/-}$ mice display reduced fat mass and resistance to high-fat diet-induced obesity.
To study the physiological role of GRASP55, the Grasp55 gene in mouse was inactivated by replacing all exons with a neomycin expression cassette (Fig. 1a;

Supplementary Fig. 1a, b) and the absence of GRASP55 protein in multiple tissues was confirmed by immunoblotting (Fig. 1b). Mice lacking Grasp55 (Grasp55$^{-/-}$) did not exhibit significant deviation from the expected genotype distribution at birth (Supplementary Table 1), or in survival at up to 40 weeks of age (Fig. 1c). However, Grasp55$^{-/-}$ mice showed no weight gain after approximately 12 weeks of age when fed on a normal diet (Fig. 1d, e; Supplementary Fig. 1c, d). Examination of relative organ weights revealed that white adipose tissue (Supplementary Table 2) and body fat mass (Fig. 1f) were profoundly reduced in Grasp55$^{-/-}$ mice. Moreover, the organ weight and cell size of white (Fig. 1g–j) and brown (Supplementary Fig. 2) adipose tissues were decreased because of collapsed cellular lipid depots. Plasma levels of lipids, such as triglycerides (TGs) and total cholesterols, were significantly reduced in Grasp55$^{-/-}$ mice without affecting plasma protein levels (Supplementary Table 3). Thus, the absence of GRASP55 in mice caused defects in body lipid accumulation. Of note, Grasp55$^{-/-}$ mice displayed resistance to a high-fat (60 cal% of fat, composition shown in Supplementary Table 4) diet-induced weight gain (Fig. 1e) and body fat mass increase (Fig. 1f). Moreover, gross and microscopic analyses of epididymal white adipose tissue (EWAT) indicated that GRASP55 deficiency profoundly inhibited the high-fat diet-induced EWAT mass increase (Fig. 1g, h) and fat cell hypertrophy (Fig. 1i, j).

Grasp55 deletion did not alter total energy intake (Fig. 2a), but reduced total energy expenditure (Fig. 2b). Considering the reduced energy expenditure and normal dietary food intake, the low body weights of Grasp55$^{-/-}$ mice (Fig. 1e) imply that Grasp55$^{-/-}$ mice may suffer from impaired energy absorption. Interestingly, the reduced energy expenditure of Grasp55$^{-/-}$ mice was significantly restored in mice fed the high-fat diet (Fig. 2b, inset). A recent report suggested that deletion of dGrasp affected adipocyte lipid storage in fly due to defects in dGRASP-mediated unconventional secretion of Upd2, a functional homolog of human leptin, which resulted in decreased Upd2-mediated insulin secretion[18]. Therefore, we measured plasma leptin and insulin levels in Grasp55$^{-/-}$ mice. Plasma leptin levels were significantly reduced in Grasp55$^{-/-}$ mice under both fasting and postprandial conditions (Fig. 2c). However, when the postprandial leptin levels were normalized to body fat mass (Fig. 1f), leptin levels were comparable between wild-type (WT, Grasp55$^{+/+}$) and Grasp55$^{-/-}$ mice (Fig. 2d; see the "Discussion" section). Plasma insulin levels were reduced in Grasp55$^{-/-}$ mice (Fig. 2e), whereas plasma glucose levels were not significantly altered by Grasp55 ablation (Fig. 2f). Furthermore, results of an insulin tolerance test indicated that insulin sensitivity of Grasp55$^{-/-}$ mice was significantly increased (Fig. 2g). Collectively, these results imply that Grasp55 deletion alters plasma levels of leptin and insulin secondarily to reduction in body fat mass and improved insulin sensitivity, suggesting that mechanisms other than reduced secretion of leptin and insulin would be primarily responsible for the lipid storage defects in Grasp55$^{-/-}$ mice.

### GRASP55 is required for intestinal fat absorption and chylomicron secretion.
An important clue to understand the mechanism of the lipid storage defect observed in Grasp55$^{-/-}$ mice was obtained via morphological examination of the intestines. In contrast to that of adipocytes, lipid storage of intestinal epithelial cells was markedly increased in Grasp55$^{-/-}$ mice, especially after olive oil feeding (Fig. 3a–d), showing supersized LDs in electron-microscopic (EM) images (Fig. 3e, f). These results imply that intestinal dietary lipid handling is differently affected by Grasp55 deletion.

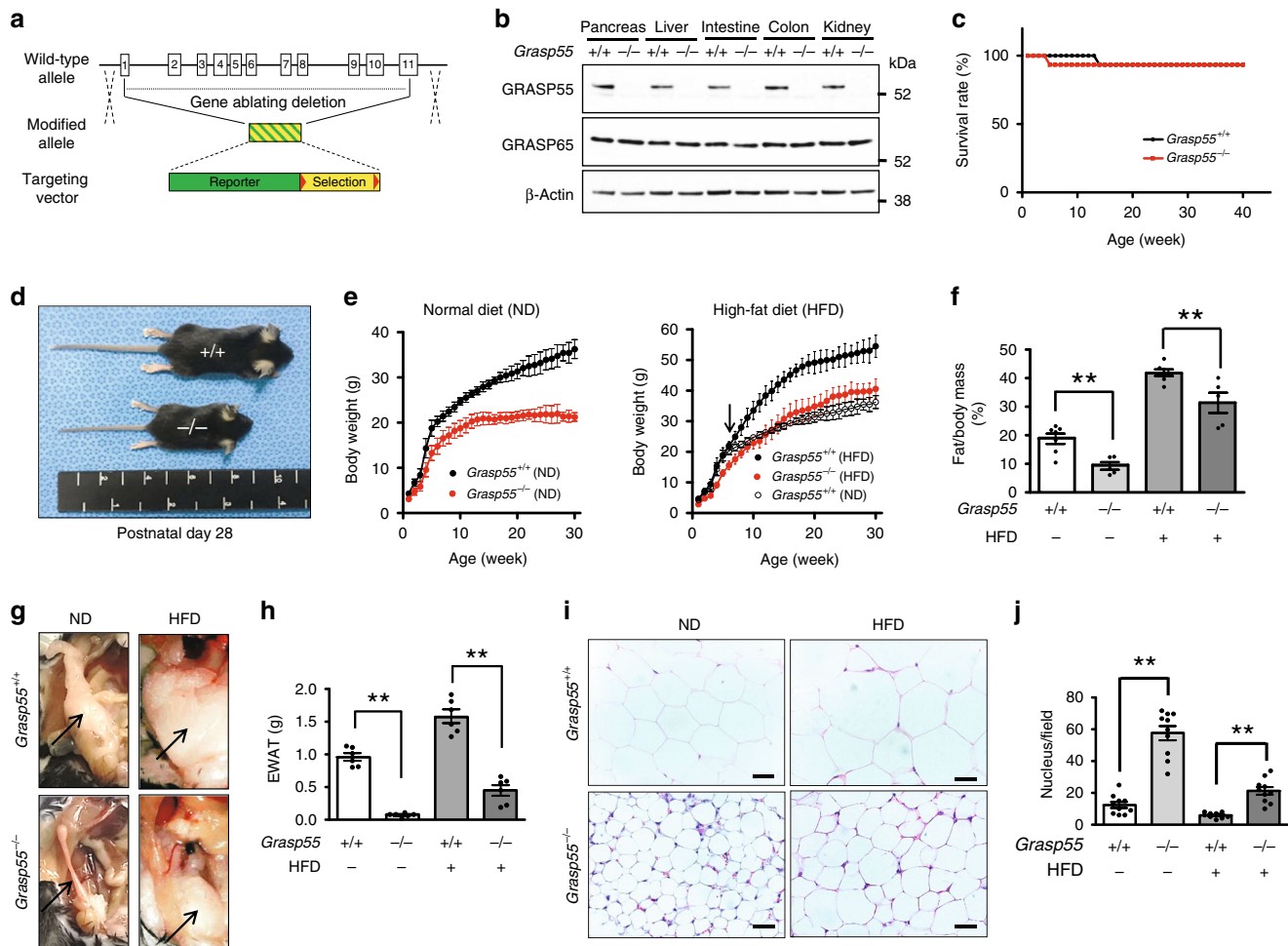

**Fig. 1 *Grasp55⁻/⁻* mice display reduced fat mass and resistance to high-fat diet-induced obesity. a** Knockout strategy for the *Grasp55* locus. The reporter-neo cassette replaced all 11 exons of mouse *Grasp55*. Red triangles indicate the location of *loxP* sites. More detailed diagrams are provided in the Supplementary Fig. 1. **b** Expressions of GRASP55 and GRASP65 in major organs were examined by immunoblotting. The amount of β-actin protein was monitored as a cytosolic protein loading control. **c** Survival curves of wild-type (*Grasp55⁺/⁺*) and *Grasp55⁻/⁻* male mice over 40 weeks (n = 15). **d** Photographs of *Grasp55⁺/⁺* and *Grasp55⁻/⁻* mice at postnatal day 28. **e** Body weight curves of male mice fed normal diet (ND, left) and high-fat diet (HFD, right) at 1–30 weeks of age (n = 15). The arrow indicates the initiation time of HFD at 6 weeks of age. The body weight curve of the *Grasp55⁺/⁺* mice fed ND was replotted in the HFD graph (open circle, right) for comparison. **f** The relative fat mass ratio (fat/body weight) of 16-week-old male mice fed ND and after HFD for 4 weeks (n = 5–7). **g–j** Gross and histological examinations of epididymal white adipose tissue (EWAT). Ventral view (**g**) and wet weights (**h**) of EWAT (right side) of male mice at 18 weeks of age fed on ND and after HFD for 12 weeks (n = 6). Arrows indicate epididymal fat tissues. Representative light microscopic tissue images (H&E staining) are presented in (**i**). Quantitative analyses of the H&E images, that inversely correlate with fat contents of EWAT by counting the number of adipocyte nuclei in a light microscopic field (450 × 340 μm), are depicted in (**j**, n = 10). Unprocessed blots are presented in Supplementary Fig. 22. Data are shown as mean ± SEM. Scale bars: 50 μm. **p < 0.01. All p values were calculated by unpaired two-tailed Student's t tests. Source data are provided as a Source Data file.

Dietary lipids, principally TGs, are hydrolyzed within the small intestine to fatty acids (FAs) and 2-monoacylglycerol (MAG) by intraluminal lipases. After uptake across the apical membrane, FAs and MAG are re-esterified into TGs by acyltransferases in enterocytes[8]. Enterocytes then complete the absorption process by secreting the lipids across the basolateral membrane, predominantly in the form of chylomicrons, which are the main contributors to the postprandial serum lipid levels[19]. Interestingly, ultrastructural analyses indicated that deficiency of GRASP55 led to fewer chylomicrons in intestinal epithelial cells prepared 2 h after oil bolus (Fig. 3g, h), suggesting reduced chylomicron secretion. A chylomicron is composed of a core of neutral lipids, mostly TGs with traces of cholesteryl ester, and a shell of amphipathic lipids of phospholipids and cholesterol. They are coated with and stabilized by the structural protein apolipoprotein B (APOB)[8]. Examination of APOB blood levels

revealed that APOB levels were significantly reduced in *Grasp55⁻/⁻* mice without affecting its protein levels in the jejunum (Supplementary Fig. 3), further supporting the possibility of reduced chylomicron secretion in the absence of GRASP55.

We then examined the effects of GRASP55 deficiency on intestinal lipid absorption. The oral fat tolerance test in the presence of tyloxapol, which inhibits lipoprotein catabolism, showed that dietary TG absorption was strongly reduced in *Grasp55⁻/⁻* mice (Fig. 4a), implying that GRASP55 depletion inhibits chylomicron secretion from the intestines. An oral fat tolerance test in the absence of tyloxapol also indicated that GRASP55 deficiency evoked reductions in postprandial plasma TG levels (Supplementary Fig. 4a). However, there were no significant differences in the absorption of amino acids between WT and *Grasp55⁻/⁻* mice (Supplementary Fig. 4b–d), indicating that lipid absorption is selectively reduced by GRASP55

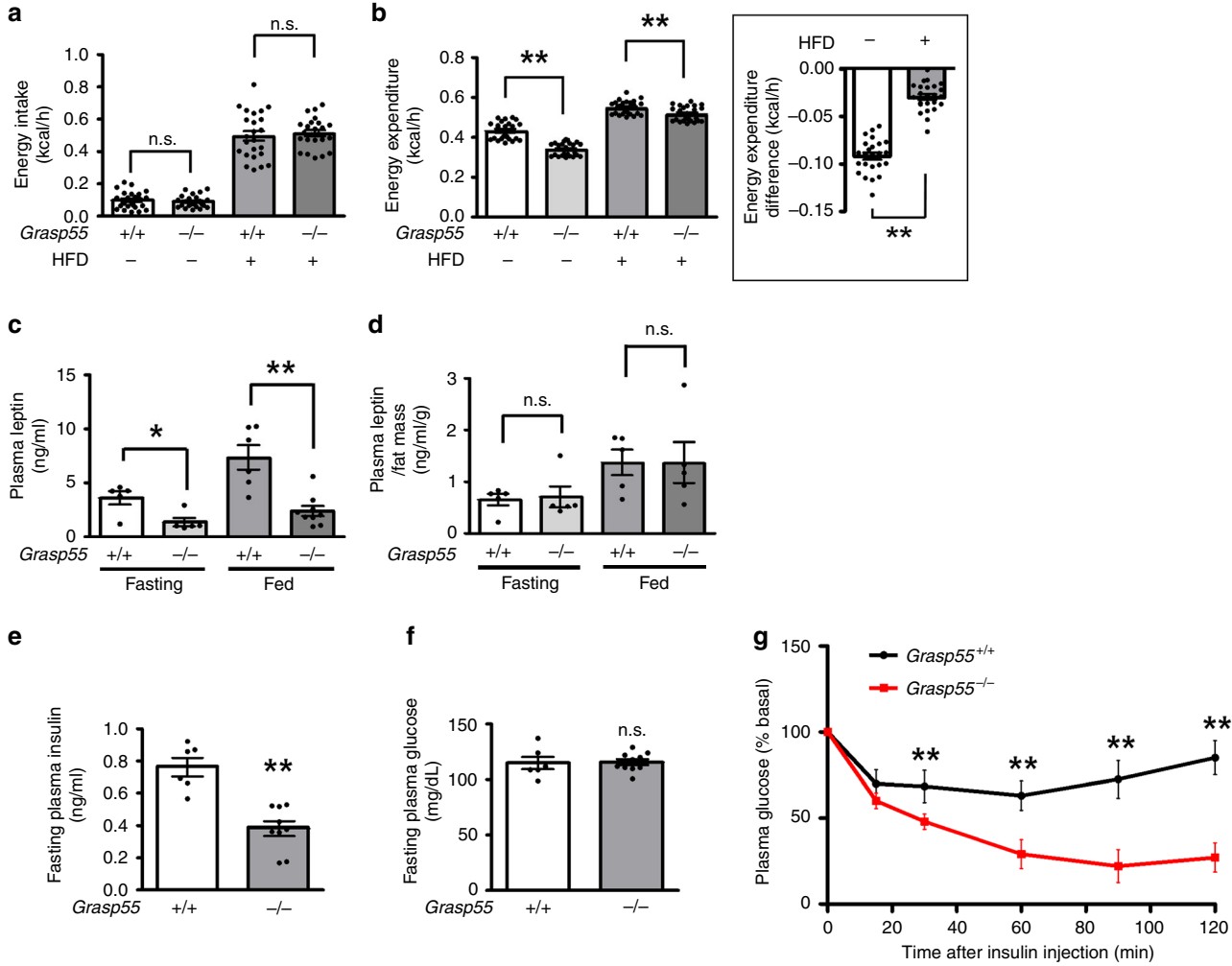

**Fig. 2 Metabolic profiles of *Grasp55*⁻/⁻ mice. a, b** Analyses of energy intake (**a**) and energy expenditure (**b**) were conducted in 12-week-old *Grasp55*+/+ and *Grasp55*⁻/⁻ mice fed normal diet (ND) or high-fat diet (HFD). Energy expenditure differences between *Grasp55*+/+ and *Grasp55*⁻/⁻ mice are plotted in (**b** inset). Data are shown as average values from a 2-day observation (*n* = 6). **c, d** Plasma leptin levels of 12-week-old *Grasp55*+/+ and *Grasp55*⁻/⁻ mice that were fasted for 16 h and fed ND (*n* = 6). Plasma leptin levels normalized to body fat mass of mice are presented in **d** (*n* = 6). **e, f** Fasting plasma levels of insulin (**e**) and glucose (**f**) were measured in 12-week-old *Grasp55*+/+ and *Grasp55*⁻/⁻ mice fasted for 16 h (*n* = 6). **g** An insulin tolerance test was performed in mice fasted for 4 h. The relative ratios of blood glucose levels to basal glucose levels were determined at the indicated time points following intraperitoneal administration of human insulin (0.75 U kg⁻¹ body weight) in 12-week-old *Grasp55*+/+ and *Grasp55*⁻/⁻ mice (*n* = 6). Data are shown as mean ± SEM. n.s.: not significant, *$p < 0.05$, **$p < 0.01$. All *p* values were calculated by unpaired two-tailed Student's *t* tests. Source data are provided as a Source Data file.

deficiency. The decrease in lipid absorption of *Grasp55*⁻/⁻ mice was also identifiable by reduced turbidity of blood plasma lipid suspensions collected 2 h after intragastric olive oil administration (Fig. 4b). Notably, fractionation analysis of plasma lipids revealed decreased TG (Fig. 4c, d) and cholesterol (Supplementary Fig. 5) content in the chylomicron/VLDL fraction of blood samples from *Grasp55*⁻/⁻ mice obtained 2 h after the oil bolus, further suggesting that GRASP55 depletion inhibits chylomicron secretion. In addition, in vitro experiments with differentiated Caco-2 cells of human intestine origin demonstrated that GRASP55 depletion inhibited transepithelial lipid absorption (shGRASP55; Fig. 4e–g). Notably, these defects were rescued by exogenous GRASP55 supplementation (GRASP55-Myc; Fig. 4e–g).

**GRASP55 participates in LD targeting of ATGL and MGL.** We next explored the mechanisms that might underlie the reduced chylomicron secretion and increased LD size caused by GRASP55 deficiency in the mouse jejunum. GRASP55 participates in Golgi stacking and ribbon linking as an adhesive peripheral Golgi

membrane protein[20]. We therefore investigated whether structural abnormalities in the Golgi cisternae were responsible for the impairment of lipid transport. Although some Golgi membranes appeared to have lost their characteristic flattened cisternae, the GRASP55 deficiency alone did not cause Golgi unstacking (Supplementary Fig. 6), which is consistent with a previous in vitro analysis[21]. In addition, the deficiency of GRASP55 did not affect protein expression levels (Supplementary Fig. 7a, b) or perinuclear Golgi localization of other Golgi cisternal adhesive proteins (Supplementary Fig. 7c), such as GM130, Golgin97, GRASP65, and Golgin45, in the mouse jejunum.

To investigate whether GRASP55 deficiency affects the intracellular trafficking of proteins involved in lipid metabolism, we examined changes in the subcellular localization of apolipoproteins (APOB, APOA1, APOA4), regulatory proteins known to be involved in assembly of chylomicrons (ARF1, ARFRP1, CIDEB, GBF1, MTP)[22–24], and LD-associated proteins and lipases (ADPN, ATGL, CGI58, CES1, CES3, AADAC, MGAT2, DGAT2, HSL, LAL, LIPC, MGL)[19,25,26]. Of great interest, some

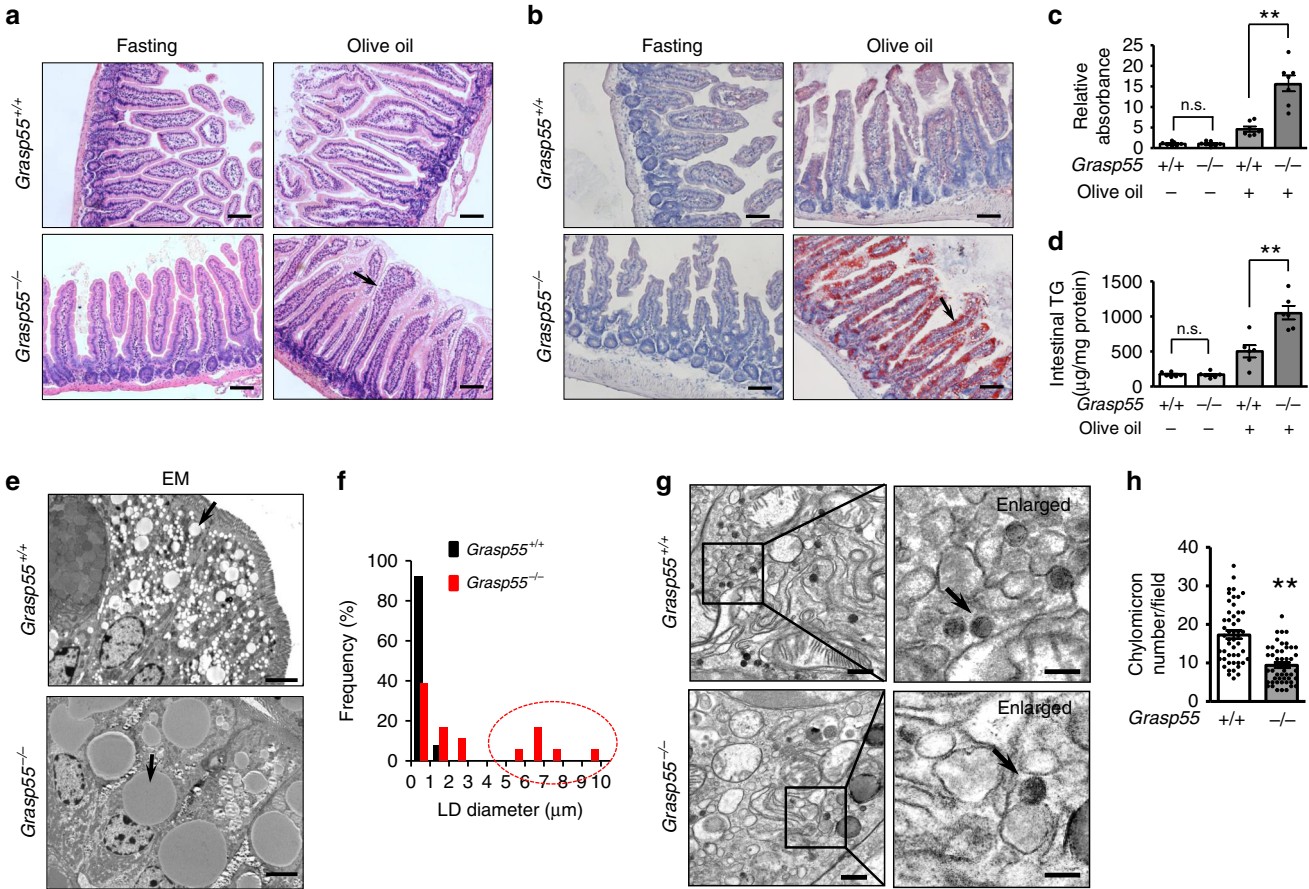

**Fig. 3 GRASP55 deficiency causes supersized LDs in intestinal epithelium. a–c** H&E and Oil Red O staining images of jejunal epithelia. Intestinal tissues were taken from 12-week-old $Grasp55^{+/+}$ and $Grasp55^{-/-}$ mice that were fasted for 16 h (**a**) or 4 h after oral gavage of olive oil (**b**, 10 µl g$^{-1}$ of body weight). Quantification of stained lipid droplets (LDs) was performed by measuring sample absorbances at 510 nm (**c**, $n = 7$). The readings were normalized to the background value obtained from a non-induced fasting control. Scale bars: 50 µm. **d** Intestinal triglyceride (TG) contents were measured in jejunums of 12-week-old $Grasp55^{+/+}$ and $Grasp55^{-/-}$ mice that were fasted for 16 or 4 h after olive oil bolus (olive oil, 10 µl g$^{-1}$ of body weight, $n = 6$). **e** Electron-microscopic (EM) images of mouse jejunum 4 h after oral gavage of olive oil (10 µl g$^{-1}$ of body weight). Arrows indicate cytosolic LDs. Quantitative analyses of LD diameters are summarized in (**f**) ($Grasp55^{+/+}$, $n = 313$ from five mice; $Grasp55^{-/-}$, $n = 307$ from five mice). Dashed circle represents the supersized LD (diameter, >5 µm). GRASP55 deficiency increased the LD size from 0.74 ± 0.02 µm to 3.19 ± 0.16 µm ($p < 0.01$). Scale bars: 5 µm. **g**, **h** EM images showing chylomicrons of mouse intestinal epithelial cells. Mouse jejunums were prepared 2 h after oral gavage of olive oil. Arrows indicate chylomicron particles. Quantitative analyses of chylomicrons are summarized in (**h**). The number of chylomicron particles in each EM field (3.72 × 3.72 µm) of the Golgi region was determined ($n = 50$ from five mice each). Scale bars: 500 nm (enlarged: 200 nm). Data are shown as mean ± SEM. n.s.: not significant, **$p < 0.01$. All $p$ values were calculated by unpaired two-tailed Student's $t$ tests. Source data are provided as a Source Data file.

LD-associated lipases, such as ATGL and MGL, were identified in the Golgi fraction and their contents in the Golgi were markedly reduced in the jejunum of $Grasp55^{-/-}$ mice (Fig. 5a; Supplementary Fig. 8). The presence of ATGL (Supplementary Fig. 9) and MGL (Supplementary Fig. 10) in the Golgi was also confirmed by immunostaining. Notably, deficiency of GRASP55 diminished colocalization of the LD-associated lipases and the Golgi marker protein GM130, which was more marked under lipid-rich conditions (after olive oil feeding; Supplementary Figs. 9, 10). Moreover, the defects in Golgi localization were associated with reductions in the total amount of ATGL (Fig. 5b, c; Supplementary Fig. 11a–d) and MGL (Supplementary Fig. 11e, f) proteins in the $Grasp55^{-/-}$ jejunum. However, mRNA levels of the genes encoding these proteins were not affected by GRASP55 deficiency (Supplementary Fig. 11g, h).

LDs in enterocytes are believed to be a transient pool of stored lipids that provide sustained lipid supply for chylomicron assembly and secretion after a meal[19]. Because ATGL regulates LD lipid content[25] and its protein amount was more significantly reduced by GRASP55 deficiency than MGL (Fig. 5b, c;

Supplementary Fig. 11e, f), we further investigated the role of GRASP55 in the intracellular regulation of ATGL as a model for the Golgi-traveling LD proteins. Notably, the Golgi localization defects of ATGL appeared to cause failure of LD targeting, and thus the amount of ATGL in LD (Fig. 5d, e) and colocalizations between ATGL and the LD marker protein ADRP (Fig. 5f, g) were profoundly reduced in the jejunums of $Grasp55^{-/-}$ mice. The LD-targeting failure of ATGL upon GRASP55 depletion was also evident in Caco-2 (Fig. 6a–c) and HeLa cells (Supplementary Fig. 12) using short-hairpin RNAs (shRNAs) against GRASP55 and staining LDs with BODIPY dye. In vitro analyses in Caco-2 cells revealed that GRASP55 depletion resulted in decreased stability of ATGL (Fig. 6d, e). Of note, these LD localization and protein stability defects of ATGL caused by GRASP55 depletion were rescued by supplementation with exogenous GRASP55 (GRASP55-Myc) in Caco-2 (Fig. 6c–e) and HeLa cells (Supplementary Fig. 12). Furthermore, a high level of exogenous ATGL (ATGL-Myc) supplementation, which restored LD-associated ATGL levels in GRASP55-depleted cells, rescued the transepithelial lipid absorption defects in Caco-2 cells (Fig. 6f, g). Expression

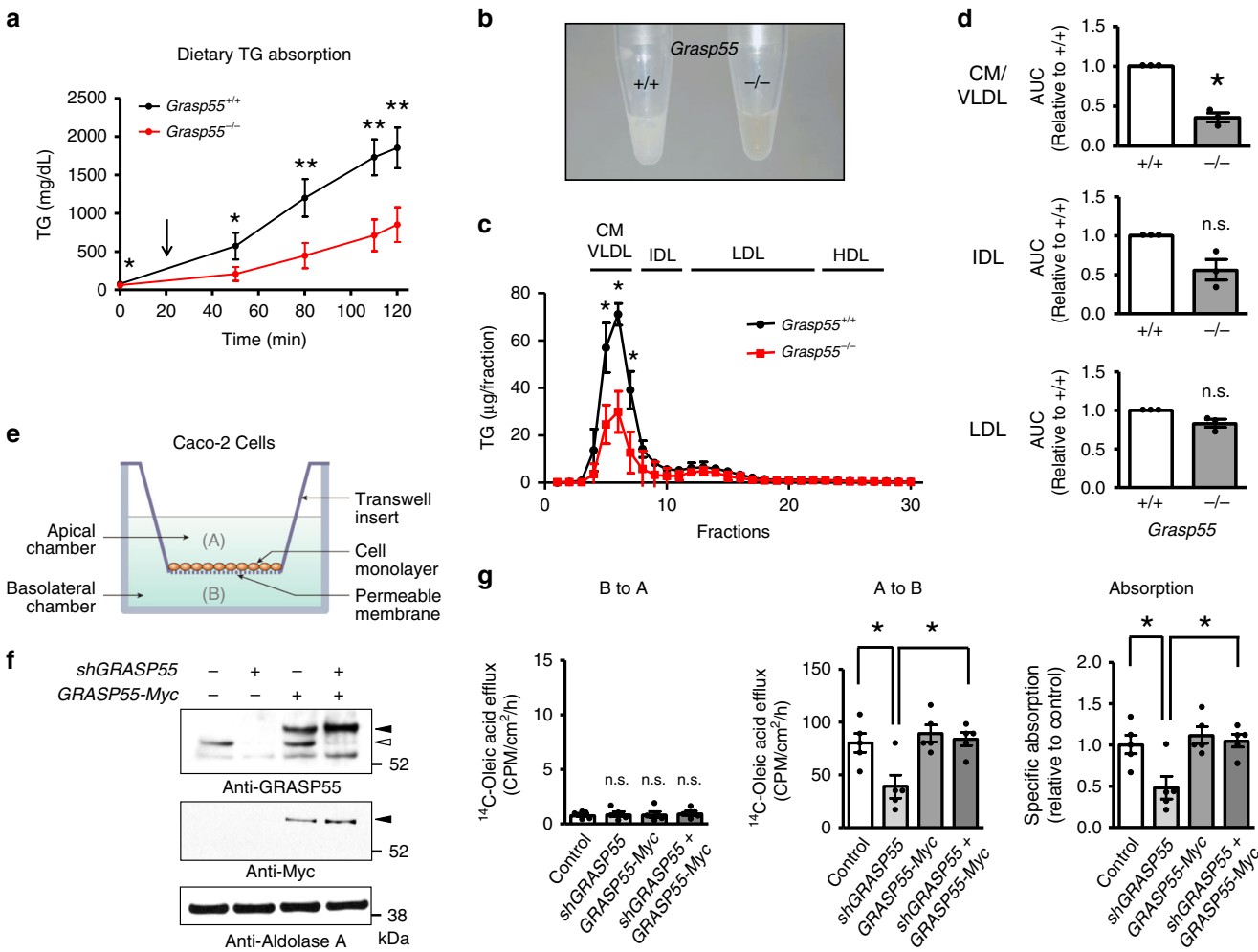

**Fig. 4 GRASP55 is required for intestinal fat absorption and chylomicron secretion. a** An oral fat tolerance test was performed with 12-week-old male mice after oral gavage of olive oil (10 μl g$^{-1}$ of body weight). Tyloxapol (Triton WR-1339) was administered 20 min after the olive oil gavage (arrow, 0.5 g kg$^{-1}$ intravenous injection) to inhibit lipoprotein catabolism. Plasma triglyceride (TG) concentrations were measured at the indicated time points ($n = 6$). **b** Photograph of blood plasma taken from WT and *Grasp55$^{-/-}$* male mice 2 h after application of olive oil. **c, d** Fractionation analysis of plasma lipids from mice using fast performance liquid chromatography. Blood samples were taken 2 h after oral garage of olive oil. TG concentrations were determined in each fraction. The area under curve (AUC) of plasma lipoprotein fractions were measured using ImageJ software and summarized in (**d**, $n = 3$ independent experiments from pooled blood plasma samples of 3–4 mice). **e** Schematic diagram of the in vitro two-chamber system with differentiated Caco-2 cells. **f** Immunoblot results showed that the short-hairpin RNA against *GRASP55* (*shGRASP55*) specifically depleted endogenous GRASP55 protein expression and GRASP55 with a COOH-terminal Myc-tag (*GRASP55-Myc*) induced exogenous GRASP55 protein. The filled arrowhead and the open arrowhead indicate levels of exogenous GRASP55 and endogenous GRASP55, respectively. Aldolase A was monitored as a cytosolic protein loading control. **g** Transepithelial lipid transport of Caco-2 monolayers were determined using $^{14}$C-labeled oleic acid. The majority of the exogenous oleic acids were incorporated into TGs and secreted as TGs in Caco-2 cells[47]. Basolateral (B) to apical (A) transport of lipids (left, B to A) was much smaller than that of apical to basolateral transport (middle, A to B), and was not affected by GRASP55 depletion or overexpression. Specific transepithelial lipid absorption was determined by subtracting values of (B to A) from those of (A to B) in each paired experiment (right, $n = 5$). GRASP55 depletion reduced lipid absorption by 49.2 ± 14.5%. GRASP55 overexpression rescued the lipid absorption defect in GRASP55-depleted Caco-2 cells. CPM, counts per min. CM, chylomicron, VLDL, very-low-density lipoprotein; IDL, intermediate density lipoprotein; LDL, low density lipoprotein; HDL, high density lipoprotein. Unprocessed blots can be found in Supplementary Fig. 22. Data are shown as mean ± SEM. n.s.: not significant, *$p < 0.05$, **$p < 0.01$. P values were calculated by unpaired (**a, c**), paired (**d**) two-tailed Student's *t* tests or ANOVA followed by Tukey's multiple comparison tests (**g**). Source data are provided as a Source Data file.

of GRASP55-Myc alone did not significantly affect the LD targeting and protein stability of ATGL in control cells (Fig. 6c–e; Supplementary Fig. 12).

Immunoprecipitation assays in Caco-2 cells showed that GRASP55 reciprocally associated with ATGL, which was further augmented under lipid-rich conditions that was mostly due to increased total cellular amounts of ATGL (Fig. 7a–d). However, GRASP55 did not localize on LDs even under lipid-rich conditions (Fig. 5a; Supplementary Fig. 13), suggesting that GRASP55 interacts with ATGL primarily at the Golgi, where both

proteins are enriched (Fig. 5a). Notably, pull-down assays using protein fragments of GRASP55 and ATGL indicated that the GRASP55-PDZ1 domain directly interacts with the ATGL-patatin domain (Fig. 7e, f). The ATGL-patatin domain has been shown to interact with many other regulatory proteins of ATGL, such as CGI58, HILPDA, and G0S2[27]. Altogether, the above results suggest that GRASP55 participates in Golgi localization of ATGL via physical association.

To gain mechanistic insight into how GRASP55 regulates ATGL trafficking and stability, we analyzed ATGL levels in the

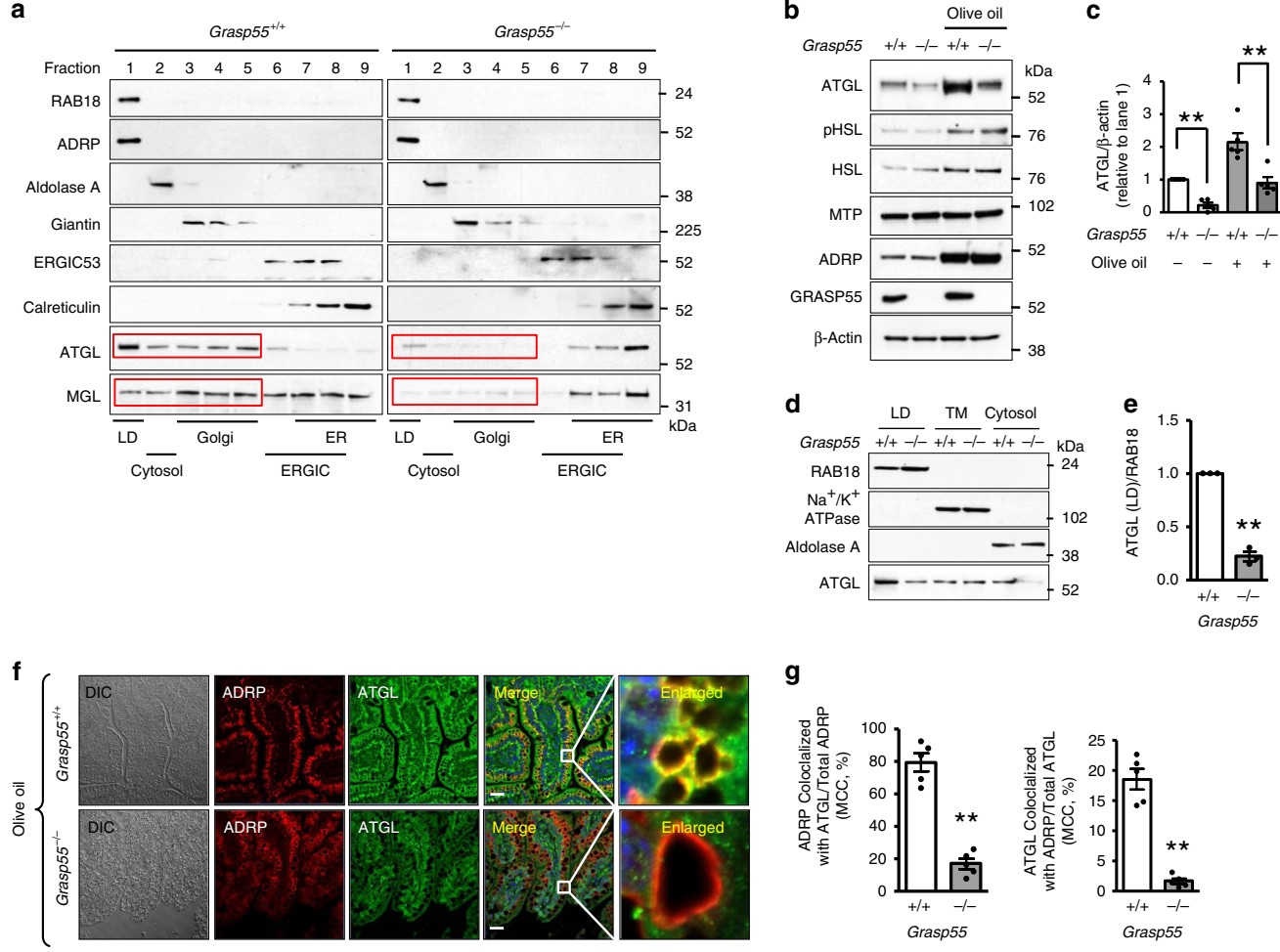

**Fig. 5 GRASP55 participates in the Golgi localization and LD targeting of ATGL and MGL. a** Subcellular localization of ATGL and MGL in the intestinal epithelial cells of *Grasp55*$^{+/+}$ and *Grasp55*$^{-/-}$ mice was determined using LD floating ultracentrifugation (for Fraction 1), followed by subcellular fractionation assay (for Fractions 2–9; using pooled jejunum tissue from four male mice 4 h after olive oil bolus [olive oil, 10 µl g$^{-1}$ of body weight]) as described in Methods. RAB18/ADRP, Aldolase A, Giantin, ERGIC53, and calreticulin were used as organelle markers of the LDs, cytosol, Golgi, the ER-Golgi intermediate compartment (ERGIC), and the ER, respectively. Immunoblotting results with a full list of proteins involved in lipid metabolism are shown in the Supplementary Fig. 8. The Golgi localizations of ATGL and MGL were reduced by GRASP55 deficiency in mouse intestinal cells (red boxes). Three independent experiments showed similar results. **b, c** The protein expressions of ATGL, phospho-HSL (pHSL), HSL, MTP, and ADRP were analyzed by immunoblotting. Jejunum tissues were prepared from *Grasp55*$^{+/+}$ and *Grasp55*$^{-/-}$ mice fasted for 16 or 4 h after olive oil bolus (olive oil, 10 µl g$^{-1}$ body weight). Representative immunoblots are shown in (**b**). Densitometric analysis of ATGL are shown in (**c**, *n* = 5). A summary of pHSL, HSL, MTP, and ADRP is shown in Supplementary Fig. 11a–d. The level of β-actin was monitored as a cytosolic protein loading control. **d, e** Measurements of ATGL in the lipid droplet (LD) fraction of intestinal epithelial cells (using pooled jejunum tissue from four male mice 4 h after olive oil bolus [olive oil, 10 µl g$^{-1}$ of body weight]). Representative immunoblots are shown in (**d**) and a summary of multiple experiments is shown in (**e**, *n* = 3). TM, total membrane. **f, g** Intracellular localization of ATGL and ADRP, a marker protein of LD, in jejunal epithelia of mice 4 h after olive oil bolus. Representative images are shown in (**f**) and quantitative analyses of colocalization between ATGL and ADRP are presented in (**g**, *n* = 5). MCC, Manders' colocalization coefficient. Unprocessed blots can be found in Supplementary Fig. 22. Data are shown as mean ± SEM. Scale bars: 20 µm. **p < 0.01. P values were calculated by unpaired (**c**, **g**) or paired (**e**) two-tailed Student's *t* tests. Source data are provided as a Source Data file.

cytosol and LDs after treatment with the proteasomal inhibitor MG132. Interestingly, MG132 reversed the decreased cytosolic levels of ATGL caused by GRASP55 depletion (Fig. 8a, b), which was consistent with a previous report showing that non-LD-associated ATGL was sensitive to proteasomal degradation[28]. However, MG132 did not significantly affect LD-associated ATGL levels that had been reduced in GRASP55-depleted cells (Fig. 8c, d). In contrast, the add-back of GRASP55 restored ATGL levels in both the cytosol and LDs (Fig. 8a–d). We then analyzed the association between GRASP55 and ATGL after blockade of ER-to-Golgi trafficking by dominant-inhibitory forms of SAR1 (SAR1-T39N) and ARF1 (ARF1-T31N). SAR1 is a GTPase that plays a critical role in the COPII-mediated conventional ER-to-Golgi

transport[29]. Although ARF1 is known to play a central role in COPI-mediated retrograde Golgi-to-ER transport, its inhibition by dominant-inhibitory ARF1 mutants also blocks anterograde ER-to-Golgi transport[29]. Notably, the interaction between GRASP55 and ATGL was abolished by the ER-to-Golgi blockade, suggesting that transport of ATGL to the Golgi is required for ATGL to associate with GRASP55 (Fig. 8e, f). As reported earlier[28], SAR1 and ARF1 inhibition reduced ATGL levels in the cytosol and LDs (Supplementary Fig. 14). While MG132 recovered the cytosolic ATGL levels that were reduced by the ER-to-Golgi blockade (Supplementary Fig. 14), it did not recover the defects in the GRASP55-ATGL association (Fig. 8e, f) or the reductions in LD-associated ATGL induced by SAR1 and ARF1 mutants

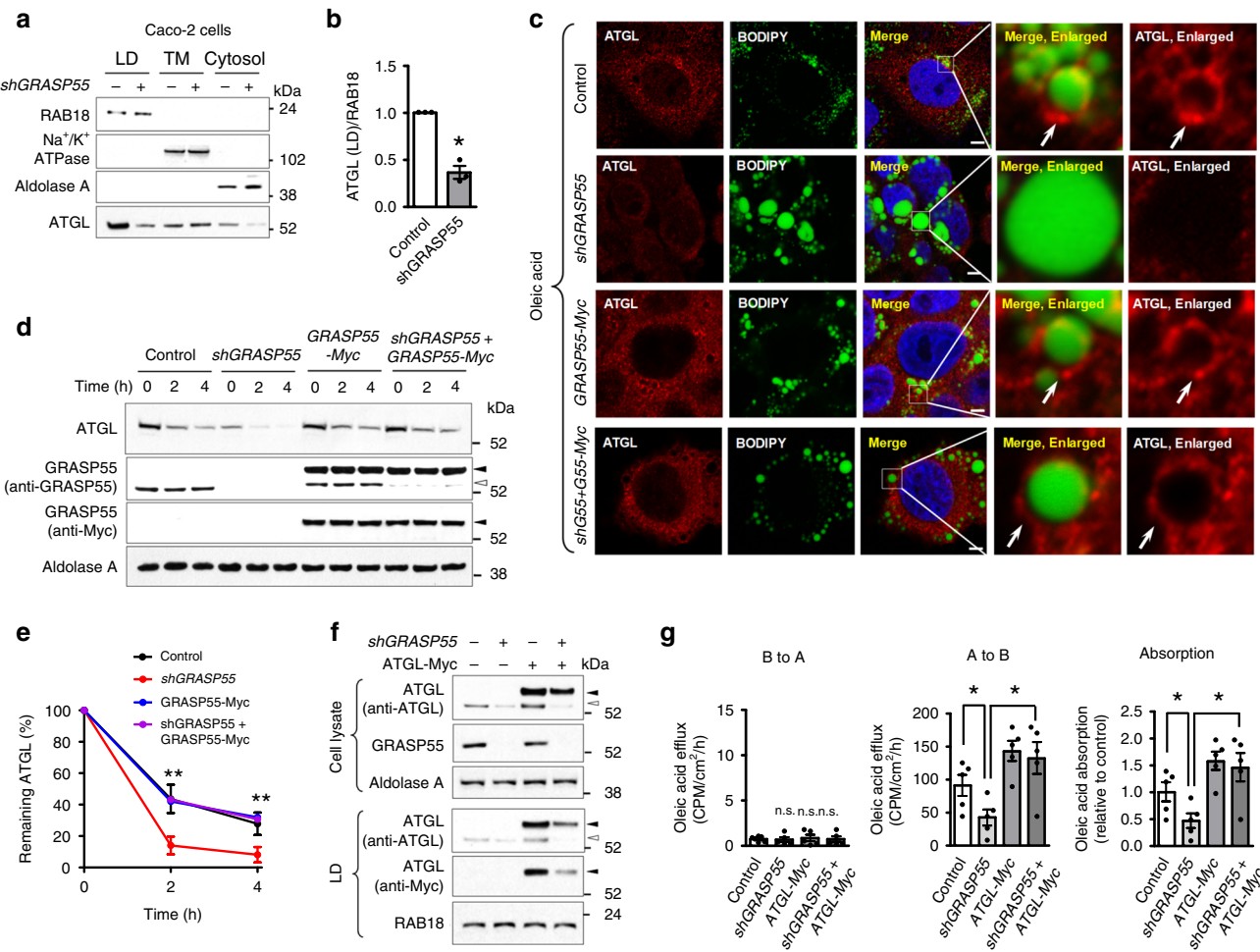

**Fig. 6 GRASP55 depletion reduces LD targeting of ATGL and transepithelial lipid absorption in Caco-2 cells. a, b** Measurements of ATGL in the lipid droplet (LD) fraction of Caco-2 cells after treatment with 400 µM oleic acid for 16 h. Representative immunoblots are shown in (**a**) and a summary of multiple experiments is shown in (**b**, n = 3). **c** Presence of ATGL on the LD surface was examined by co-staining with anti-ATGL antibodies and BODIPY 493/503 (an LD marker) in Caco-2 cells, in which endogenous GRASP55 was depleted (*shGRASP55, shG55*) and/or exogenously supplemented (*GRASP55-Myc, G55-Myc*). Cells were treated with 400 µM oleic acid for 4 h before immunostaining. Arrows indicate the location of ATGL, which is targeted to the peripheral regions of LDs. Four independent experiments showed similar results. **d, e** ATGL protein stability was examined in Caco-2 cells, in which GRASP55 was depleted (*shGRASP55*, open arrowhead) and/or exogenously supplemented (*GRASP55-Myc*, filled arrowhead). Cytosolic proteins were collected 0, 2, and 4 h after treatment with cycloheximide (0.1 mg ml$^{-1}$), an inhibitor of protein biosynthesis. Aldolase A was used as a cytosolic protein loading control. Representative immunoblot is shown in (**d**), and the results of multiple experiments are summarized in (**e**, n = 3). **f** Immunoblot results of Caco-2 monolayers, in which GRASP55 was depleted (*shGRASP55*) and exogenous ATGL (*ATGL-Myc*) was supplemented. Levels of endogenous ATGL (open arrowhead) and exogenous ATGL (filled arrowhead) are shown. A high level of ATGL-Myc supplementation restored LD-associated ATGL levels in GRASP55-depleted cells, although the LD transport efficiency was much lower (compare endogenous and exogenous ATGL levels in lanes 3 and 4). **g** Transepithelial lipid transport of Caco-2 monolayers were determined using $^{14}$C-labeled oleic acid. The basolateral (B) to apical (A) transport of oleic acid (left, B to A) was much smaller than that of apical to basolateral transport (middle, A to B), and it was not affected by GRASP55 depletion or ATGL overexpression. Transepithelial lipid absorption was determined by subtracting values of (B to A) from those of (A to B) in each paired experiment (right, n = 4). ATGL overexpression rescued lipid absorption in GRASP55-deficient Caco-2 Cells. Unprocessed blots are presented in Supplementary Fig. 22. CPM, counts per min. TM, total membrane. Data are shown as mean ± SEM. Scale bars: 10 µm. n.s.: not significant, *p < 0.05. P values were caluculated by paired (**b**), unpaired (**e**) two-tailed Student's t tests or ANOVA followed by Tukey's multiple comparison tests (**g**). Source data are provided as a Source Data file.

(Supplementary Fig. 14). In aggregate, these results imply that defective LD localization of ATGL is the primary event caused by GRASP55 depletion, and that the restoration of cytosolic ATGL levels to near control levels is not sufficient to recover the LD trafficking defect of ATGL.

Cellular localizations (Supplementary Fig. 8) and levels (Fig. 5b; Supplementary Fig. 11a–d) of other LD-related proteins in the mouse jejunum, such as HSL, MTP, and ADRP, were not altered. Recent reports suggested that some Golgi proteins, such as ARF1, ARFRP1, and GBF1, are involved in the regulation of LD and chylomicron formation[23,30]; however, GRASP55 depletion did not

affect their cellular localizations or levels in mice (Supplementary Fig. 8) and Caco-2 cells (Supplementary Fig. 15). It has been shown that GRASP proteins are involved in autophagy[31,32], which is known to play a role in LD dynamics in enterocytes[33]. To analyze the effects of autophagy, we examined the conversion of microtubule-associated protein light chain 3 (LC3) into its phosphatidylethanolamine-conjugated form (LC3-II) as a parameter of autophagic flux. As shown in Supplementary Fig. 16, *Grasp55* deletion did not significantly affect autophagic flux in the mouse jejunum, suggesting that altered autophagy is not responsible for the formation of supersized LDs in *Grasp55$^{-/-}$* mice.

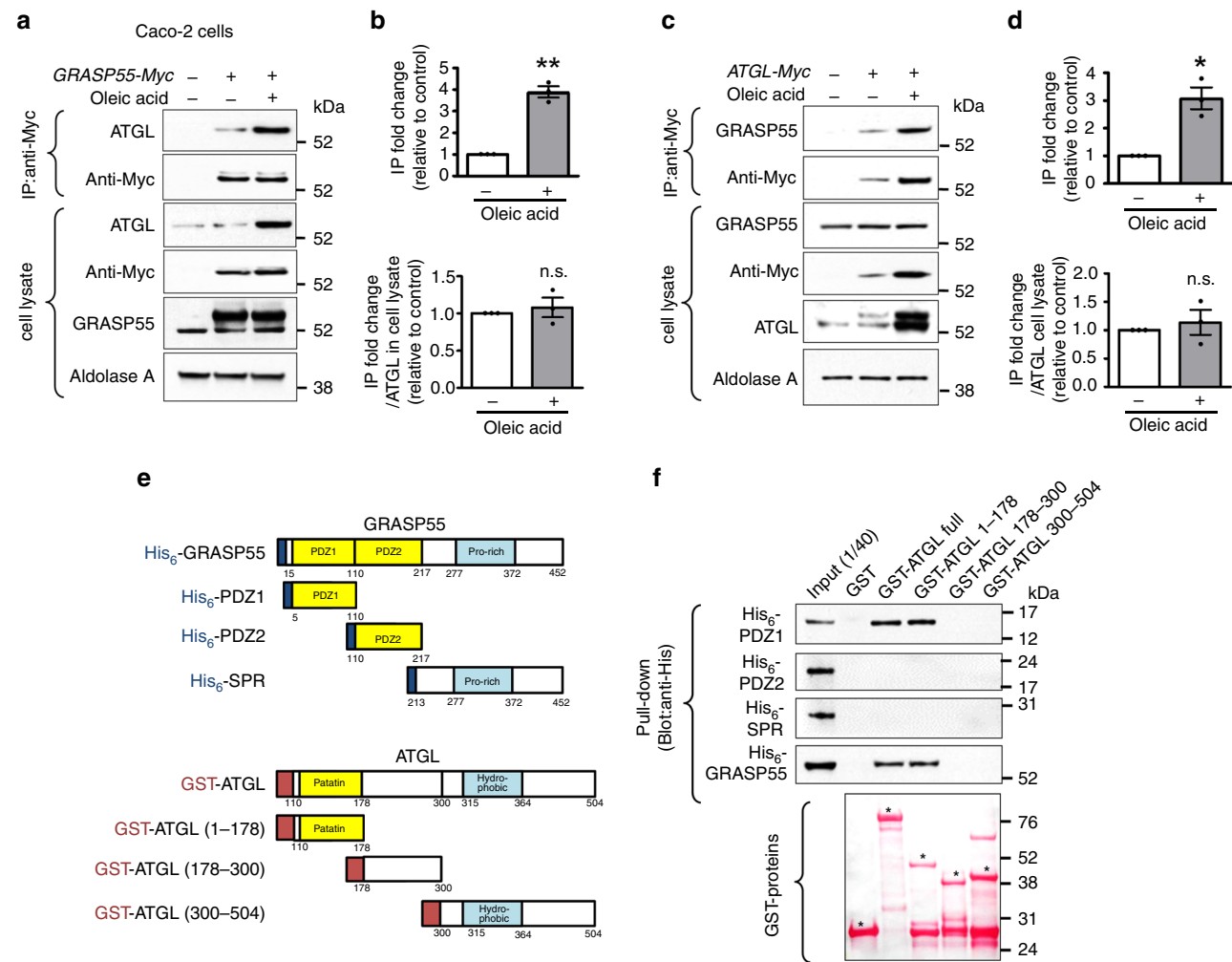

**Fig. 7 GRASP55-PDZ1 interacts with ATGL-patatin. a–d** Coimmunoprecipitation experiments with anti-Myc antibodies were performed in Caco-2 cells. For the induction of ATGL protein expression, some cells were treated with 400 μM oleic acid for 16 h. A representative coimmunoprecipitation assay using GRASP55 with a COOH-terminal Myc-tag (*GRASP55-Myc*) is shown in (**a**), and the results of multiple experiments are summarized in (**b**, $n = 3$). A representative coimmunoprecipitation assay using ATGL with a COOH-terminal Myc-tag (*ATGL-Myc*) is shown in (**c**), and the results of multiple experiments are summarized in (**d**) ($n = 3$). Aldolase A was used as a cytosolic protein loading control. **e, f** Pull-down assays were performed with GRASP55 and ATGL fragments. The domain structures of His$_6$-tagged GRASP55 and GST-tagged ATGL constructs used in this study are shown in (**e**), and a representative result from the pull-down assay is shown in (**f**). Expression of control GST and each GST-fusion protein is visualized by Ponceau S staining (**f** lowermost panel). An asterisk indicates the band of each GST-tagged protein. One microgram of each His$_6$-tagged protein was loaded as an input control. The PDZ1 domain of GRASP55 interacted strongly with the ATGL-patatin domain. Three independent experiments showed similar results. Unprocessed blots can be found in Supplementary Fig. 22. Data are shown as mean ± SEM. n.s.: not significant, *$p < 0.05$, **$p < 0.01$. All $p$ values were calculated by paired two-tailed Student's $t$ tests. Source data are provided as a Source Data file.

The decrease in ATGL protein levels by GRASP55 deficiency was also observed in the mouse liver (Supplementary Fig. 17a–d). As an LD-associated lipase, reduced ATGL levels were expected to cause an enlarged LD. However, in contrast to those of the jejunum, LD size (Supplementary Fig. 17e, f) and TG contents (Supplementary Fig. 17g) of hepatocytes were reduced. Taken together with those in adipocytes (Fig. 1g–j; Supplementary Fig. 2), reduced lipid contents in hepatocytes imply that impaired intestinal fat absorption by GRASP55 deficiency dominantly affects the organismal level of lipid handling in mice.

**Lipid accumulation defects in *Grasp55*$^{-/-}$ mice are rescued by the supplementation of GRASP55.** Next, we generated mice overexpressing GRASP55 with a COOH-terminal Myc-tag (*TgGrasp55*) and crossed them with *Grasp55*$^{-/-}$ mice to examine whether the lipid accumulation defect phenotype of *Grasp55*$^{-/-}$ mice is rescued by GRASP55 supplementation. The body size, weight, and lipid storage of control *TgGrasp55* mice on the WT background were comparable to those of WT mice (Fig. 9a–f). Immunostaining with the Golgi marker protein GM130 revealed that transgenic GRASP55-Myc proteins were mostly localized on the Golgi apparatus in jejunal epithelial cells (Supplementary Fig. 18). Importantly, transgenic expression of GRASP55 restored the growth and body weights of *Grasp55*$^{-/-}$ mice (Fig. 9a, b) and rescued the organ size and fat contents of adipose tissues (Fig. 9c–f). Furthermore, transgenic GRASP55 eliminated excessive lipid accumulation in the small intestine after intragastric olive oil application (Fig. 9g) and restored the reduced levels of ATGL (Fig. 9h, i) in the *Grasp55*$^{-/-}$ mouse jejunum. These results indicate that GRASP55 deficiency is responsible for the reduced body fat mass and intestinal lipid handling defects in

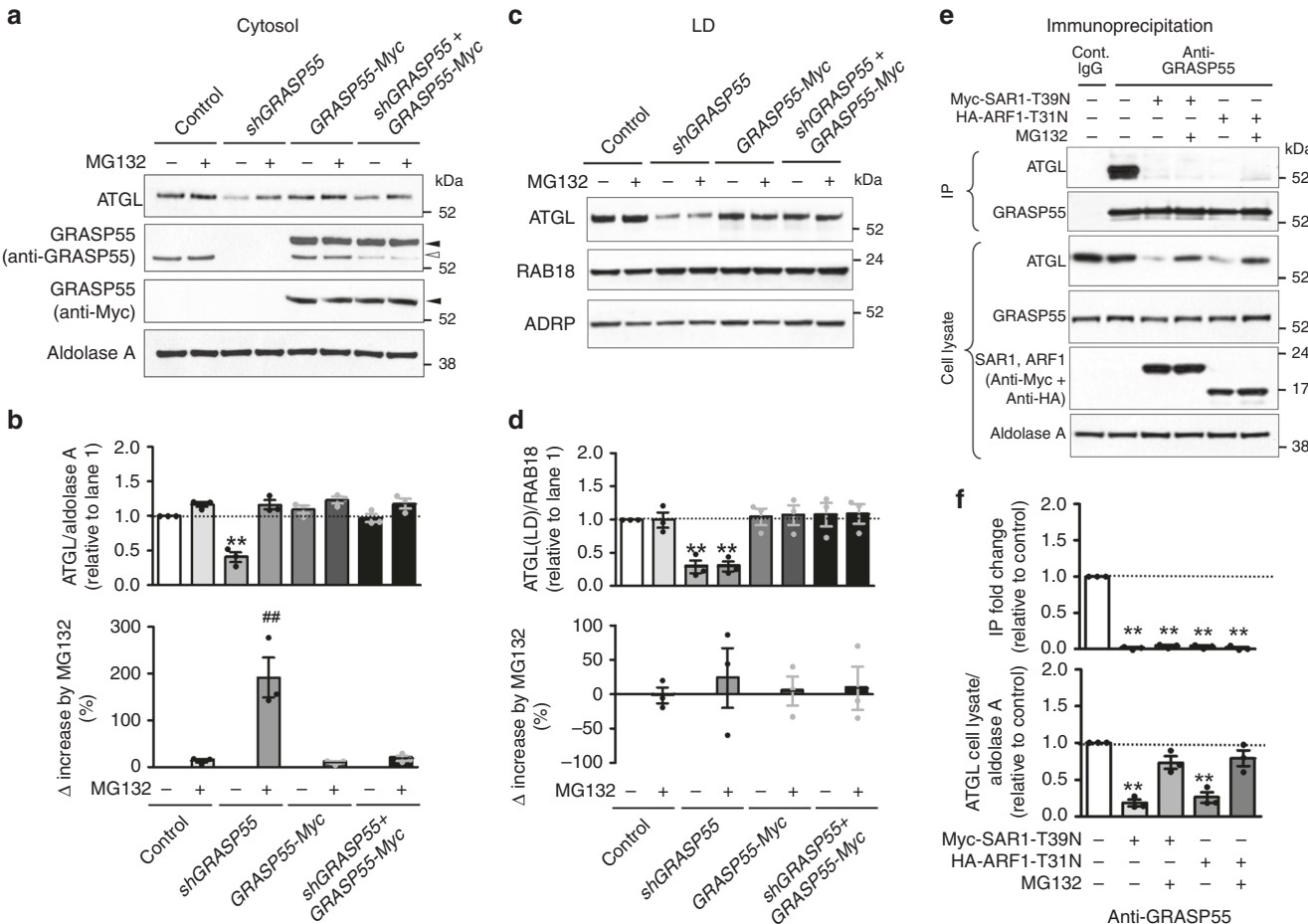

**Fig. 8 Decreased ATGL levels in LDs by GRASP55 depletion are not reversed by proteasome inhibition. a–d** Immunoblot analysis of ATGL levels in Caco-2 cells, in which GRASP55 was depleted (*shGRASP55*) and/or exogenously supplemented (*GRASP55-Myc*). Cytosolic proteins were collected in the absence or presence of the proteasomal inhibitor MG132 (0.5 μM, 16 h). For the induction of ATGL protein expression, cells were treated with 400 μM oleic acid for 16 h. Representative immunoblots of cytosols and LDs are shown in (**a**) and (**c**), and the results of multiple experiments are summarized in (**b**) and (**d**), respectively (*n* = 3 each). Aldolase A and RAB18/ADRP were used as loading controls for cytosolic and LD proteins, respectively. MG132 reversed the decreased cytosolic levels of ATGL by GRASP55 depletion, but did not significantly affect the ATGL levels in LDs. In contrast, the add-back of GRASP55 restored ATGL levels in both the cytosol and LDs. **e, f** Coimmunoprecipitation (IP) experiments with anti-GRASP55 antibodies were performed in Caco-2 cells. The ER-to-Golgi trafficking was blocked by dominant-inhibitory forms of SAR1 (SAR1-T39N) and ARF1 (ARF1-T31N). For the induction of ATGL protein expression, cells were treated with 400 μM oleic acid for 16 h. A representative IP assay is shown in (**e**), and the results of multiple experiments are summarized in (**f**, *n* = 3). Aldolase A was used as a cytosolic protein loading control. MG132 does not rescue the dissociation of GRASP55 and ATGL induced by SAR1-T39N and ARF1-T31N. Unprocessed blots can be found in Supplementary Fig. 22. Data are shown as mean ± SEM. **\*\****p* < 0.01, difference from control without MG132. **##***p* < 0.01, difference from control. All *p* values were calculated by ANOVA followed by Tukey's multiple comparison tests. Source data are provided as a Source Data file.

*Grasp55*$^{-/-}$ mice and that nonspecific or off-target effects are not involved in the altered lipid metabolism of *Grasp55*$^{-/-}$ mice.

**Lipid storage defects of *dGrasp*$^{-/-}$ are restored by the enterocyte-specific supplementation of dGRASP or BMM in *Drosophila melanogaster*.** To test whether the role of GRASP in lipid absorption is evolutionarily conserved, we examined the lipid storage phenotypes of GRASP deletion in fly. We obtained a fly strain with the *dGrasp*-null mutant allele (*dGrasp*$^{-/-}$), also known as *dGrasp*$^{302}$, harboring the deletion of the entire *dGrasp* coding region[34]. The *dGrasp*-null flies (*dGrasp*$^{-/-}$) are viable, but the females are sterile. Notably, *dGrasp*-null flies exhibited lipid accumulation in their posterior midguts after a standard diet feeding (Fig. 10a–d) and decreased whole-body TG contents (Fig. 10e), implying defective intestinal lipid absorption (control [1st lane] vs. *dGrasp*$^{-/-}$ [2nd lane], Fig. 10d, e). The introduction of *dGrasp*

cDNA in the *dGrasp*$^{-/-}$ background using *NP1-GAL4*, which specifically labels enterocytes[35], restored whole-body fat amounts and rescued the intestinal lipid accumulation defect (*dGrasp*$^{-/-}$ [2nd lane] vs. *NP1>dGrasp;dGrasp*$^{-/-}$ [4th lane], Fig. 10d, e). Enterocyte-specific knockdown of *dGrasp* (*NP1>dGrasp-i*) also evoked lipid accumulation in the fly midgut (Supplementary Fig. 19a, b). Collectively, these results suggest that the lipid handling defects of *dGrasp*$^{-/-}$ flies are due to loss of *dGrasp* in enterocytes.

Because mutation of *Grasp55* in mice caused ATGL mislocalization and decreased expression levels in the jejunum, we next sought to test whether the intestinal lipid accumulation defect of *dGrasp*$^{-/-}$ flies can be rescued by overexpression of *Drosophila* lipid storage droplet-associated TG lipase Brummer (BMM), a homolog of mammalian ATGL[17]. The increased lipid accumulation in the midgut of *dGrasp*$^{-/-}$ flies was rescued by overexpression of intestinal *bmm* using *NP1-GAL4* in the *dGrasp*$^{-/-}$ background

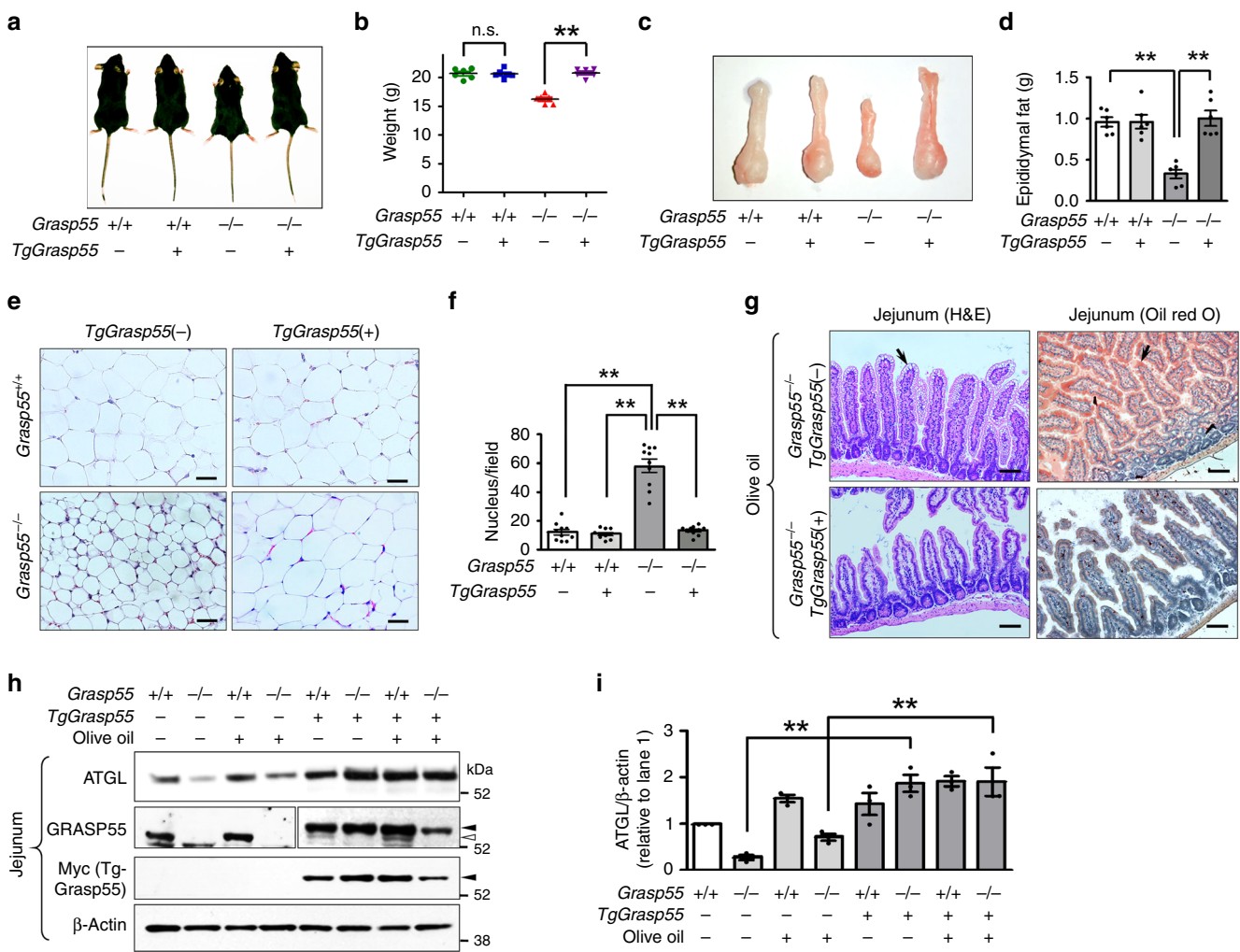

**Fig. 9 GRASP55 deficiency-induced lipid storage defects are rescued by supplementation of GRASP55 in mice.** Wild-type (*Grasp55*^+/+) and *Grasp55*^−/− mice were crossed with transgenic mice overexpressing GRASP55-Myc (*TgGrasp55*). **a**, **b** Photographs (**a**) and a summary of body weight (**b**, $n = 6$) of 12-week-old male mice. **c**, **d** Photographs (**c**) and a summary of weight (**d**, $n = 6$) of epididymal white adipose tissues (EWAT, right side). **e**, **f** Representative light microscopic images (H&E) of EWAT are shown in (**e**). Quantitative analyses of the H&E images, that inversely correlate with fat contents of EWAT by counting the number of adipocyte nuclei in a light microscopic field ($450 \times 340 \mu m$), are presented in (**f**, $n = 6$). **g** H&E and Oil Red O staining of mouse jejunum 4 h after olive oil bolus ($10 \mu l\, g^{-1}$ body weight). *TgGrasp55*(+) reduced the postprandial lipid accumulation of jejunal epithelia in *Grasp55*^−/− mice. Arrows indicate supersized LDs. **h**, **i** The protein amount ATGL of mouse jejunum was analyzed by immunoblotting. Representative immunoblots are shown in (**h**). In some experiments, jejunum tissues were prepared 4 h after olive oil bolus ($10 \mu l\, g^{-1}$ body weight). The filled arrowhead and open arrowhead indicate levels of the exogenous GRASP55-Myc and endogenous GRASP55, respectively. A blot with a longer exposure time of GRASP55 is shown in the left four lanes to better visualize the presence and absence of endogenous GRASP55. The results of multiple experiments ($n = 4$) are summarized in (**i**). *TgGrasp55*(+) increases the protein levels of ATGL in jejunal epithelia of *Grasp55*^−/− mice. Unprocessed blots can be found in Supplementary Fig. 22. Data are shown as mean ± SEM. Scale bars: 50 μm. n.s.: not significant, **$p < 0.01$. *P* values were calculated by unpaired two-tailed Student's *t* tests (**b**) or ANOVA followed by Tukey's multiple comparison tests (**d**, **f**, **i**). Source data are provided as a Source Data file.

(*dGrasp*^−/− [2nd lane] vs. *NP1>bmm;dGrasp*^−/− [6th lane], Fig. 10d). Importantly, the reduced whole-body TG levels were also restored by the intestinal BMM supplementation (*dGrasp*^−/− [2nd lane] vs. *NP1>bmm;dGrasp*^−/− [6th lane], Fig. 10e). These results indicate that reduced TG hydrolysis in LD is responsible for the GRASP deficiency-induced excessive lipid accumulation and reduced intestinal lipid absorption in the *Drosophila* midgut. Interestingly, deletion of *bmm* also induced intestinal lipid accumulation in flies; however, this defect was not reversed by *dGrasp* supplementation (Supplementary Fig. 19c, d), indicating that *bmm* is epistatic to *dGrasp* and also most likely located downstream.

## Discussion

In the present study, we have shown that the Golgi and GRASP55 functions are critically involved in lipid homeostasis, in

particular, by regulating intestinal lipid uptake. We found that intestinal chylomicron secretion and LD regulation are sensitive to Golgi defects evoked by GRASP55 deficiency. Notably, some LD-associated proteins, such as ATGL and MGL, reach their final destination via the Golgi complex. Previously, a group of reports showed that blocking the function of COPI and ARF1 evoke supersized LDs, presumably through deterioration of ATGL function (BMM in *Drosophila*) in mammalian and *Drosophila* cells[28,36]. The authors initially thought that COPI-mediated transport from the ER or the ER-Golgi intermediate compartment (ERGIC), rather than the Golgi, might be responsible for the LD targeting of ATGL. However, blocking the function of COPI and ARF1 also disrupts the Golgi both structurally and functionally[37]. Furthermore, results in the present study showed that ATGL is abundantly localized in the Golgi (Fig. 5a;

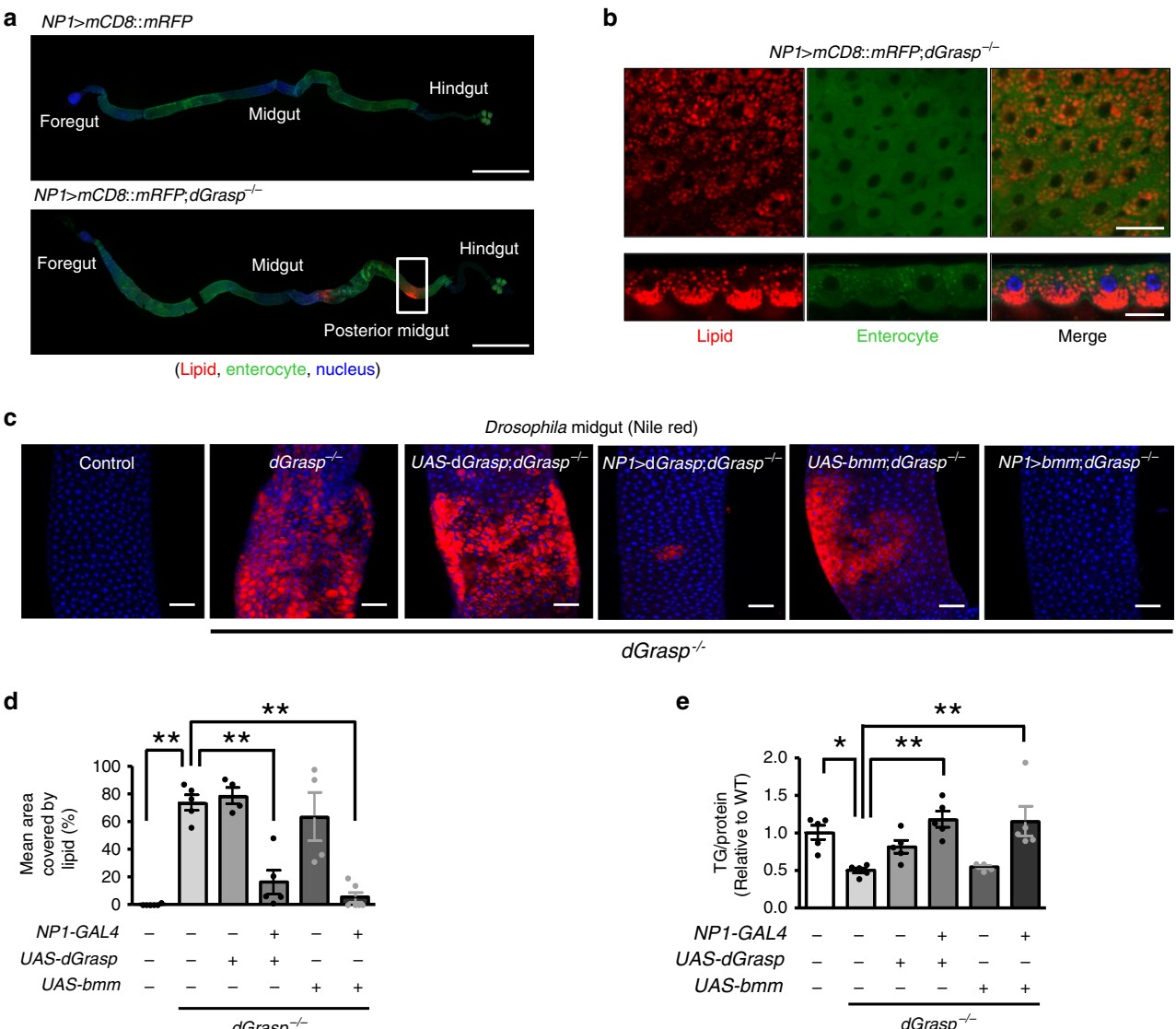

**Fig. 10 The role of dGRASP in intestinal lipid absorption and whole-body TG storage in adult flies.** The *dGrasp*-null mutant allele (*dGrasp*[−/−]), also known as *dGrasp*[302], harbors a deletion of the whole *dGrasp* coding region. **a**, **b** Lipid accumulation in enterocytes of posterior midguts of *dGrasp*[−/−] flies. Enterocytes of *NP1>mCD8::RFP* fly guts were stained with antibodies against DsRed (green) and nuclei were counterstained with TO-PRO-3 (blue). Lipids were visualized by staining with BODIPY solution (red). For consistency, pseudocolors were used. Confocal microscopic images of whole guts from control (*NP1>mCD8::RFP*) and *dGrasp*-mutant flies (*NP1>mCD8::RFP;dGrasp*[−/−]) are shown in (**a**). Scale bars: 1 mm. Boxed area indicates the posterior midgut where lipids accumulated. Image sections of the boxed area taken from the top (upper, scale bar: 25 μm) and lateral (bottom, scale bar: 10 μm) orientations are shown in (**b**). **c**, **d** Confocal microscopic images of lipid accumulation in the fly guts of the indicated genotypes. The fly genes *dGrasp* and *bmm* are homologs of mammalian *GRASP* and *ATGL*, respectively. Each gene was expressed in the enterocytes of *dGrasp*[−/−] flies under the control of NP1-GAL4. Lipids were visualized by staining with Nile Red (red) and nuclei were counterstained with TO-PRO-3 (blue). Representative images of the posterior midguts are shown in (**c**), and quantitative analyses of intestinal lipid accumulation in the indicated genotypes of the flies are presented in (**d**), n = 4-6. Scale bars: 50 μm. **e** Quantitative analyses of relative whole-body TG contents in the indicated genotypes of the flies (n = 5). UAS, upstream activation sequence. Data are shown as mean ± SEM. *p < 0.05, **p < 0.01. All p values were calculated by ANOVA followed by Tukey's multiple comparison tests. Source data are provided as a Source Data file.

Supplementary Fig. 8) and deficiency of the Golgi protein GRASP55 markedly reduces the amount of ATGL in the Golgi and LD (Fig. 5a, d, e; Supplementary Fig. 9), implying that blocked ATGL transport from the Golgi could be primarily responsible for the supersized LDs induced by COPI and ARFI inhibition[28,36]. Therefore, increased LD size by COPI and ARF1 inhibition[28,36] are compatible with the present findings of GRASP55 deficiency in the mouse jejunum (Fig. 3). However, in other tissues, such as fat tissues (Fig. 1g–j) and liver

(Supplementary Fig. 17), the same GRASP55 deficiency caused collapse of LDs, suggesting that retarded intestinal fat absorption is the dominant phenotype of global deletion of the *Grasp55* gene in vivo, and thus maintaining the Golgi-mediated chylomicron secretion would be one of the most important physiological roles of GRASP55 in live animals.

Recent evidence suggests that some Golgi proteins are involved in the lipidation of the cytosolic organelle LD. For example, two small GTPases, ARFRP1 and ARL1, and their downstream targets

on the trans-Golgi, Golgins and RAB proteins, have been shown to participate in the control of LD morphology and trafficking[38]. Based on this finding that the ARFRP1-ARL1-Golgin-RAB cascade is involved in LD lipidation, it has been proposed that LDs may travel through the Golgi[39]. While chylomicrons travel through the Golgi lumen, LDs, if they do indeed travel through the Golgi, may do so via peripheral association because they are present in the cytosolic space. However, in our extensive EM examinations, we were unable to find any Golgi structures that were peripherally associated with LD-like structures. Although it is uncertain whether LDs themselves can physically interact with and travel through the Golgi, the present results indicate that at least some LD proteins travel through the Golgi before targeting LDs. Interestingly, blockade of ER-to-Golgi transport by SAR1 and ARF1 inhibition reduced LD localization of RAB18 and ADRP as well as that of ATGL (Supplementary Fig. 14). In contrast, GRASP55 depletion inhibited LD localization of ATGL without affecting that of RAB18 and ADRP (Supplementary Fig. 14), indicating that ATGL is individually recruited to the LD by GRASP55 after traveling through the Golgi. The IP results, in which interactions between GRASP55 and ATGL were prevented by the blockade of ER-to-Golgi transport (Fig. 8e, f), further demonstrate that ATGL associates with GRASP55 at the Golgi. The GRASP55 deletion evoked an increase in the luminal width of some Golgi stacks (Supplementary Fig. 6). Although we cannot completely rule out the possibility that Golgi flatness defects contributed to reduced chylomicron secretion, such contribution appeared to be minimal because only 7.5% of the Golgi stacks showed an abnormal enlargement (>0.1 μm).

The gene supplementation and depletion experiments involving bmm in Drosophila provide mechanistic insights into how GRASP deficiency can induce the defects in intestinal lipid transport (Fig. 10), indicating that mobilization of lipids from LD-associated TGs is required for efficient transepithelial fat absorption. Epithelial cells in the Drosophila gut secrete diacylglycerol (DAG) into hemolymph (Drosophila circulation) as a major source of lipids with the help of APOB lipoproteins LPP and LTP[40,41], while mammalian enterocytes principally secrete TGs into the blood in the form of APOB-containing chylomicrons. Therefore, inhibition of the conversion of TGs to DAG by BMM depletion may directly inhibit DAG secretion into hemolymph, and thus inhibit dietary lipid absorption in Drosophila. However, the mechanism of cytosolic LD-associated TG mobilization for chylomicron secretion in mouse enterocytes appears to be more complex.

Intestinal deficiency of the ATGL cofactor CGI58 reduces postprandial lipid absorption in mice, suggesting that mobilization of LD-associated TG is also required for lipoprotein-mediated lipid secretion in mammals[42]. Interestingly, in contrast to CGI58 deficiency, intestinal ATGL deficiency did not noticeably inhibit lipid absorption in mice[43]. The reasons for the discrepancy between CGI58 and ATGL deficiencies are unclear at present. A plausible scenario is that CGI58 activates lipase(s) other than ATGL, which would constitute the salvage pathway for mobilization of LD-associated TGs; and therefore, CGI58 deficiency can inhibit intestinal fat absorption while deficiency of ATGL alone does not. Notably, deficiency of MGL, which also exhibited reduced Golgi localization and cellular expression in response to GRASP55 deficiency in the present study (Fig. 5a; Supplementary Figs. 8, 10, 11), has been shown to inhibit lipid absorption[44]. Therefore, combinatorial effects caused by mislocalization of multiple Golgi-traveling LD proteins, including ATGL and MGL, appear to be responsible for the GRASP55 deficiency-induced reductions in intestinal lipid uptake in mice. A further dissection of TG mobilization from LDs to chylomicrons, including the potential role of MGL in this pathway, may contribute to the

search for effective therapeutic target molecules against obesity-associated diseases.

A recent study reported that deletion of dGrasp in flies reduced body TG contents, similar to the results of the present study[18]. However, in that study, the fat storage defect of GRASP-deleted flies could be rescued by fat body-specific dGrasp supplementation[18], in contrast to the present observation that the fat storage defect of dGrasp-deleted flies was rescued by enterocyte-specific dGrasp supplementation (Fig. 10e). At present, the reason for the discrepancies between the two studies is unclear. A potential mechanism could be an age-dependent effect; Rajan et al. found that the dGrasp deletion effect on fat bodies was more profound with age[18]. They also suggested that Upd2 (Drosophila homolog of mammalian leptin) is secreted by a dGRASP-mediated unconventional secretion pathway in Drosophila, and thus deletion of dGrasp resulted in a reduction in Upd2 secretion in adipocytes[18]. Of note, mammalian leptin possesses a signal peptide (also known as leader sequence) that brings the cargo protein to the ER and subsequent Golgi trafficking. This contrasts with Drosophila Upd2 that does not have a signal peptide (Supplementary Fig. 20). Furthermore, mammalian leptin has two predicted disulfide bonds while Upd2 does not[18], suggesting that the ER-Golgi luminal environment is important for proper folding of mammalian leptin, which would be critical for its extracellular adipokine function. Therefore, it appears that leptin has evolved to acquire the conventional secretory route in mammalian organisms that require more sophisticated regulation of energy metabolism. In the present study, plasma leptin levels were proportional to the body fat mass and the leptin levels normalized to fat mass were not significantly reduced by GRASP55 deficiency (Fig. 2d). These results imply that reductions in body fat mass would be responsible for the decrease in the plasma leptin levels observed in Grasp55[−/−] mice (Fig. 2c), and that the unconventional secretory function of GRASP55 does not mediate leptin secretion in mice. Similarly, reductions in body fat mass appear to be responsible for the increased insulin sensitivity of Grasp55[−/−] mice (Fig. 2g) and the resultant decrease in plasma insulin levels (Fig. 2e), without affecting plasma glucose levels (Fig. 2f). A sustained reduction in plasma insulin levels may also contribute to the growth retardation, and particularly the reduced lean body masses, observed in Grasp55[−/−] mice (Supplementary Table 2), although nutritional imbalance caused by reduced fat absorption is primarily responsible for the growth retardation.

In conclusion, we used an integrated in vivo and in vitro approach to demonstrate that GRASP55 plays a key physiological role in intestinal lipid uptake (Supplementary Fig. 21). GRASP55 deficiency leads to defects in LD targeting of ATGL and MGL, which causes the formation of supersized LDs and decreased chylomicron secretion due to reduction in lipid mobilization of LDs. In addition, low protein levels of ATGL and MGL, due to increased degradation of the mistargeted proteins, would further aggravate lipid mobilization of LDs. Increasing evidence suggests that postprandial serum levels of TGs, chylomicrons, and APOB-containing chylomicron remnants are important risk factors for atherosclerosis and metabolic disorders[22,45]. Therefore, the finding that Grasp55[−/−] mice displayed reduced fat absorption and resistance to high-fat diet-induced obesity has enormous implication in the search for druggable targets against obesity-associated diseases as well as in elucidating the underpinnings of digestive physiology.

## Methods

**Generation of Grasp55[−/−] and TgGrasp55 mice.** For generation of Grasp55[−/−] mice, embryonic stem cells harboring the Grasp55 mutation (allele name: Gorasp2[tm1(KOMP)Vlcg], strain: C57BL/6NTac, coat color: black, purchased from KOMP Repository, Oakland, CA, USA) were cultured and used in blastocyst injection into

albino mice for production of chimeric mice (Macrogen Inc., Seoul, Korea). Animals with a high degree of coat color chimerism were back-crossed to C57BL/6N mice (Orient Bio Inc., Seoul, Korea) for more than six generations. Germ line transmission was observed by coat pigment and confirmed by SNP genotyping. Heterozygous Grasp55$^{+/-}$ mice were used to generate homozygous Grasp55$^{-/-}$ mice. Primers for genotyping are as follows: GRASP55_WT (endogenous) forward: 5′-TCT CTG AGA GTG TGC TTG CTT AC-3′, reverse: 5′-GGA TTT AGG AGT TTG GTT AGC TGA-3′, GRASP55_KO (neo cassette) forward: 5′-TCA TTC TCA GTA TTG TTT TGC C-3′, reverse: 5′-TCC AAA GAT GAA CCT CCC AG-3′.

For the generation of pCAG-GRASP55-Myc transgenic mice, a pRP expression vector encoding hGRASP55-Myc under the control of cytomegalovirus (CMV) enhancer fused to the chicken beta-actin promoter (CAG promoter) was obtained from VectorBuilder (www.vectorbuilder.com). The capability of this construct to allow GRASP55 expression was tested by transient transfection into HEK293 cells. The plasmid was digested with ApaLI and SspI to excise a 4-kb fragment and then microinjected into the pronucleus of fertilized eggs. Injected eggs were implanted into C57BL/6N surrogate mothers to obtain offspring and genotyping was performed from pups by PCR to identify those carrying the GRASP55-Myc transgene (Macrogen Inc.). Protein expression of exogenous GRASP55-Myc was approximately eight times higher than that of endogenous GRASP55 in the mouse jejunum (Fig. 9h). Primers for genotyping are as follows: GRASP55 forward: 5′-TCA CTC ACT GTG GAT GTG ACG-3′, reverse: 5′-GGT TAC AAA TAA AGC AAT AGC ATC AC-3′.

Mice were bred and maintained according to the Yonsei Medical Center Animal Research Requirements, and all procedures were approved by the Committee on Animal Research at Yonsei Medical Center (protocol number 2016-0106). All animal experiments were performed using male mice and their corresponding control littermates. Mice had free access to diet and water under a 12-h light/12-h dark cycle in a temperature-controlled environment. The standard chow diet contained 10 kcal% of fat (D12450B, Research Diets, New Brunswick, NJ, USA) and the high-fat diet contained 60 kcal% of fat (D12492, Research Diets).

**Cell culture, Caco-2 differentiation, LD staining, plasmids, transfection, and gene-silencing**. HEK293T cells and HeLa cells (American Type Culture Collection [ATCC], Manassas, VA, USA; tested regularly for mycoplasma) were cultured in a DMEM medium (Thermo Fisher, Waltham, MA, USA) supplemented with 10% fetal bovine serum (FBS) and penicillin (50 IU ml$^{-1}$)/streptomycin (50 μg ml$^{-1}$, all from Thermo Fisher). Caco-2 cells (ATCC; tested regularly for mycoplasma) were cultured in a DMEM medium (Thermo Fisher) supplemented with 20% FBS, penicillin (50 IU ml$^{-1}$)/streptomycin (50 μg ml$^{-1}$), 2 mM L-glutamine, and nonessential amino acids (all from Thermo Fisher). For differentiation, Caco-2 cells were maintained for 3 weeks in a transwell system with a medium change every 2 days. Tight junction formation was verified by measuring transepithelial electrical resistance. For LD staining, 400 μM oleic acid:bovine serum albumin (BSA) complex (final concentration ratio 4:1 mM; Merck-Millipore, Burlington, MA, USA) was applied in the apical chamber for 16 h before harvest. Plasmids encoding human GRASP55-Myc, HA-Arf1, and Myc-Sar1 were described previously[46]. cDNA of GRASP55-Myc was subcloned into pCMV-Myc plasmid. Mutant plasmids were generated with a PCR-based site-directed mutagenesis kit (Stratagene, La Jolla, CA, USA) according to the manufacturer's protocol. To generate stable GRASP55 knockdown, GRASP55 overexpression, and ATGL overexpression cell lines, Caco-2 and HeLa cells were transduced with lentiviral particles produced from HEK293T cells that had been transfected with the psPAX2 packing plasmid, pMD2.G envelope plasmid, and pLKO.1 short-hairpin RNA (shRNA) plasmid (control shRNA: #SHC001, GRASP55-specific shRNA: #TRCN0000129489; Yonsei Genome Center, Seoul, Korea) or Lenti ORF clone (GRASP55: #RC202946L3, ATGL: #RC205708L3; Origene, Rockville, MD, USA). Cells stably expressing the shRNA or indicated genes were selected using puromycin.

**Immunoblot and immunoprecipitation assays**. Antibodies used in this study are listed in Supplementary Table 5. For immunoblotting, cells were homogenized in lysis buffer containing 50 mM Tris pH 7.4, 1% (v/v) NP40, 150 mM NaCl, 1 mM EDTA supplemented with protease inhibitor mixture (Roche, Applied Science, Mannheim, Germany). For tissues, the lysis buffer contained additional 100 mM benzamidine and 0.3% soybean trypsin I. Protein samples were separated by SDS-polyacrylamide gel electrophoresis. The separated proteins were transferred to a nitrocellulose membrane at 250 mA of constant current for 1 h 40 min and blotted with appropriate primary and secondary antibodies. For immunoblotting of plasma APOB, 2 μl of plasma samples were separated, and transferred at 30 mA of constant current for 16 h. Protein bands were detected by enhanced chemiluminescence (ECL) and the densities of the bands were measured using imaging software (ImageJ, National Institutes of Health, USA, https://imagej.net/Welcome). For immunoprecipitation, cell lysates were mixed with anti-Myc (Sepharose Bead Conjugate, Cell Signaling), anti-GRASP55, or control IgG (Merck-Millipore) antibodies and incubated overnight at 4 °C in a lysis buffer containing 20 mM Tris-HCl (pH 7.9), 150 mM NaCl, 1% (v/v) Triton X-100, 5 mM Na$_2$EGTA, 5% glycerol, and the complete protease inhibitor mixture (Roche Applied Science, Mannheim, Germany). Immune complexes were centrifuged at 4 °C and then washed four times with lysis buffer before elution with 2× sodium dodecyl sulfate (SDS) sample

buffer and immunoblotted. MG132 and oleic acids were purchased from Merck-Millipore.

**Lipid droplet isolation and subcellular fractionation**. The intestinal epithelia of four male mice 4 h after oral gavage of oil bolus (olive oil, 10 μl g$^{-1}$ body weight, Merck-Millipore) were homogenized with a Dounce homogenizer (Merck-Millipore). The postnuclear supernatant (PNS) was collected after centrifugation of the homogenates at 3000 × g for 10 min and was transferred into a centrifugation tube for the SW41 rotor (Beckman Coulter, Brea, CA, USA). Washing buffer (2 ml) was loaded on top of the PNS. The top LD fractions were separated by centrifugation using an Optima TLX ultracentrifuge (Beckman Coulter) with an SW41 swinging bucket rotor at 182,000 × g for 1 h. For simultaneous analyses of proteins in LDs and other organelles (Fig. 5a), remaining fractions were subjected to subcellular fractionation assays as described below.

Subcellular fractionation using OptiPrep Density Gradient Medium (Merck-Millipore) was performed as follows. Mouse jejunum tissue was washed with PBS, chopped into tiny pieces in cell suspension medium containing 0.85% (w/v) NaCl and 10 mM tricine-NaOH (pH 7.4), and homogenized by repeated passage through a 30-G syringe needle. PNS was collected after centrifugation of the homogenates at 1500 × g for 10 min and were loaded on top of the density gradient solution containing 2.5–30.0% (w/v) OptiPrep in cell suspension medium. Subcellular organelles were separated by centrifugation using an Optima TLX ultracentrifuge (Beckman Coulter) with a SW41 swinging bucket rotor at 200,000 × g for 2.5 h. Nine fractions were collected from the top to the bottom, and a 40-μl aliquot of each fraction was used for immunoblotting.

**Immunocytochemical, immunohistochemical, and microscope imaging analyses**. For antibodies see Supplementary Table 5. For immunocytochemical staining, cells were cultured on 18-mm round coverslips and fixed with 10% formalin for 10 min at room temperature. The fixed cells were washed three times with PBS and then were permeabilized with PBS containing 0.1% Triton X-100 (PBST) for 15 min at room temperature. Cells were incubated with blocking solution containing 5% donkey serum, 1% BSA, and 0.1% gelatin in PBS for 30 min at room temperature. After blocking, cells were stained by incubating with the appropriate primary antibodies overnight at 4 °C. After being washed three times with PBS, cells were then treated with fluorophore-dye conjugated secondary antibodies. For LD staining, cells were also stained with BODIPY 493/503 (Merck-Millipore). Fluorescence images were captured using a laser scanning confocal microscope with a ×63 1.4 numerical aperture (NA) oil objective lens (LSM 780, Carl Zeiss, Berlin, Germany).

For immunohistochemical analysis of intestinal tissues, jejunum was isolated, fixed in 10% neutral-buffered formalin (Merck-Millipore), and embedded in a paraffin block. Paraffin sections of the jejunum were cut, and antigen retrieval was performed by heat treatment. Nonspecific binding sites were blocked with 5% donkey serum, 1% BSA, and 0.1% gelatin in PBS for 30 min. The indicated primary antibodies were applied overnight at 4 °C. Tissues were treated with fluorophore-conjugated secondary antibodies. Fluorescence images were captured using a laser scanning confocal microscope with a ×40 1.4 NA water objective lens (LSM 780, Carl Zeiss). To stain the lipid components, paraffin slide of the jejunum was stained with Oil Red O and Mayer's hematoxylin. Microscopic images were taken using an upright microscope BX53 equipped with a Digital Camera DP73 unit (Olympus Co., Shinjuku, Tokyo, Japan).

**Pull-down assay**. The antibodies used in this study are listed in Supplementary Table 5. All recombinant fusion proteins were produced in the BL-21 (DE3) Escherichia coli strain. The synthesis of GST-fusion and His$_6$-fusion proteins was induced by the addition of 0.5 mM isopropyl β-D-1-thiogalactopyranoside at 37 °C. Recombinant proteins were subsequently purified with glutathione Sepharose beads (GE Healthcare Bio-Science AB, Björkgatan, Uppsala, Sweden) or with a nickel-nitrilotriacetic acid (Ni-NTA) protein purification system (QIAGEN, Hilden, Germany) according to the manufacturer's instructions. Eluted His$_6$-fusion proteins were mixed with 20 μg of GST-fusion recombinant proteins bound to glutathione Sepharose. Following overnight incubation at 4 °C, bead-bound complexes were washed, eluted in SDS sample buffer, and immunoblotted.

**Transmission electron microscopy**. Jejunum and liver dissected from mice were immediately fixed for 12 h with 2% glutaraldehyde–paraformaldehyde in 0.1 M phosphate (pH 7.4) at 4 °C. Samples were rinsed in 0.1 M phosphate buffer, post-fixed with 1% OsO$_4$, dehydrated in ascending concentrations of ethanol and propylene oxide, and embedded into Poly/Bed 812 kit (Polysciences, Warrington, PA, USA). After pure fresh resin embedding and polymerization at 65 °C, samples were placed in the electron-microscope oven (TD-700, DOSAKA, Japan) for 24 h. Initially, thick sections of 200–250 nm were cut and stained with toluidine blue (Merck-Millipore) for light microscopy. Ultrathin sections (~70 nm) were obtained with an ultramicrotome (UltraCut-UCT, Leica, Austria), which were then collected on 100-mesh copper grids. After staining with 2% uranyl acetate (15 min) and lead citrate (5 min), the thin sections were examined by transmission electron microscopy at 120 kV (Tecnai G$_2$ Spirit Twin, Thermo Fisher). Diameter of LD and the Golgi cisternae were analyzed using ImageJ software package as reported

previously[21]. The summarized results of LD diameter (Fig. 3f) and maximum luminal width of Golgi cisternae (Supplementary Fig. 6b) are presented using frequency distribution histograms.

**Mouse plasma parameter assays.** WT and *Grasp55*$^{-/-}$ mice were fasted for 16 h and blood was collected by cardiac puncture. Blood samples were centrifuged at $1000 \times g$ for 10 min at 4 °C. After centrifugation, 10 µl of the top yellow plasma layer was transferred into Dry Slide Reagents (Fujifilm Co., Akasaka, Minato, Tokyo, Japan). Several plasma parameters including triglycerides, total cholesterol, albumin, total bilirubin, and total protein (Supplementary Table 3) were measured by using an automated clinical chemistry analyzer, Fuji Dri-chem Nx500 (Fujifilm Co.) according to the manufacturer's instructions. Before each measurement, a QC card (Fujifilm Co.) was used to adjust the lot variability in each of the slide reagents.

**Lipoprotein analysis by fast performance liquid chromatography (FPLC).** The blood from four male mice of each group was collected by cardiac puncture 2 h after oral oil application. A sample (250 µl) of pooled plasma was subjected to gel filtration with FPLC for lipoprotein profile analyses. GE Healthcare ÄKTAexplorer equipped with a Superose 6 10/300 GL column (GE Healthcare Bio-Science AB, Björkgatan, Uppsala, Sweden) was employed with 10 mM sodium phosphate, 0.15 M sodium chloride, 0.01% (w/v) sodium EDTA, and 0.02% (w/v) sodium azide, pH 7.2 at room temperature. The FPLC system was run with a constant flow of 500 µl min$^{-1}$, and fractionation was collected after 5 min with 400 µl per fraction. The concentrations of triglyceride and total cholesterol in fractions 1–30 were determined according to the manufacturer's instruction using Infinity Triglycerides Reagent or Cholesterol Reagent from Thermo Fisher Scientific Inc. (Middletown, VA, USA).

**Mouse in vivo studies.** Metabolic assay: Fat and lean body mass were assessed using an $^1$H TD-NMR minispec system (LF90 II, Bruker Optik, Ettlingen, Germany) before and after 4 weeks on HFD. Energy balance, including food consumption and energy expenditure, was evaluated using a metabolic monitoring system (CLAMS; Columbus Instruments, Columbus, USA) for 4 days (2 days of acclimation followed by 2 days of measurement) before and after 4 weeks on HFD.

Leptin and insulin assay: To measure plasma leptin levels, mice were fasted for 16 h or allowed free access to food. Blood samples were collected via cardiac puncture, and leptin concentrations were determined using an immunoassay kit (Cayman Chemical Company, Ann Arbor, MI, USA). For the insulin tolerance test, mice were fasted for 4 h. Then, blood samples of 12-week-old male mice were collected via the tail vein prior to (basal, time 0) as well as after intraperitoneal injection of insulin (0.75 U kg$^{-1}$ body weight, Merck-Millipore) at 15, 30, 60, 90, and 120 min. Plasma glucose concentrations were measured with a glucometer (Accu-Chek, Roche Diagnostics Corp., IN, USA).

Oral fat tolerance test: For the oral fat tolerance test, mice were fasted for 16 h and then given 10 µl g$^{-1}$ of body weight olive oil by oral gavage. In some experiments, Triton WR-1339 (15% in saline, 0.5 g kg$^{-1}$; Merck-Millipore) was injected intravenously into the lateral tail vein 20 min after the olive oil gavage to inhibit lipoprotein catabolism (Fig. 4a). Blood samples (30 µl) were collected via tail veins prior to (basal, time 0) as well as 1, 2, 3, and 4 h after oral oil administration. Blood samples were then centrifuged at $1000 \times g$ for 10 min at 4 °C. After centrifugation, triglycerides and total cholesterol concentrations of plasma samples were measured using a colorimetric TG assay kit (Cayman Chemical Company) and a cholesterol quantitation assay kit (Abcam, Cambridge, UK), respectively.

Amino acid absorption: WT and *Grasp55*$^{-/-}$ mice were fasted for 16 h and then given 10 µl/g of body weight solution containing amino acids dissolved in PBS (pH 7.4) at a final concentration 10-fold higher than the plasma concentration in WT animals by oral gavage. The solution was supplemented with 1.5 µCi ml$^{-1}$ $^{14}$C-radiolabeled glycine, L-proline, or L-tryptophan (Moravek, California, CA, USA). After 1 h, animals were sacrificed, blood was collected by cardiac puncture, and the amount of radioactivity was determined.

Mouse intestinal lipid assay: To measure intestinal triglyceride and cholesterol levels, jejunum from WT and *Grasp55*$^{-/-}$ mice were collected. Lipids were extracted by the Folch extraction method[43]. The lipid extract was dried under a stream of nitrogen, and 1% Triton X-100 in chloroform was added. Lipids were dried again under a stream of nitrogen gas, and the samples were dissolved in 100 µl $H_2O$. Triglycerides, total cholesterol and cholesteryl ester concentrations were measured using a colorimetric triglyceride assay kit (Cayman) and a cholesterol quantitation assay kit (Abcam).

In vitro lipid uptake assay: [1-$^{14}$C]-radiolabeled oleic acid (55 mCi mmol$^{-1}$) and non-labeled oleic acid were purchased from Moravek and Merck-Millipore, respectively. Radiolabeled and non-labeled oleic acids were conjugated to BSA (Merck-Millipore) in serum-free DMEM, by addition of concentrated fatty acid in ethanolic stock such that the amount of ethanol was <0.5% (v/v). The molar ratio of radiolabeled oleic acid, non-labeled oleic acid, and BSA was 4 µM:4 mM:1 mM. Media was incubated for 1 h at 37 °C before addition to Caco-2 cells. Differentiated Caco-2 cell monolayers were washed twice with serum-free DMEM before adding 1 ml fatty acid-BSA complexes to the apical cell surface or 2 ml to the basolateral

side. In the opposite chamber, serum-free DMEM without BSA or fatty acid was applied. After incubation for 24 h at 37 °C, apical and basolateral media were collected and centrifuged at $1000 \times g$ for 5 min to remove cell debris. The radioactivity in the media was analyzed by using Tri-Carb 2910TR Liquid Scintillation Analyzer (Perkin Elmer, Waltham, MA, USA) and was normalized to background radioactivity.

Quantitative real-time PCR (qRT-PCR): The intestinal epithelia of male mice fasted for 16 h were transferred to RNase-free round-bottomed tube and lysed by using lysis buffer (Thermo Fisher) with 2-mercaptoethanol. After centrifugation at $12,000 \times g$ for 2 min at room temperature, the supernatant was transferred to a clean RNase-free tube and homogenized by passing 10 times through a 21-G needle attached to an RNase-free syringe. The supernatant was transferred to a new RNase-free tube after centrifugation at $12,000 \times g$ for 2 min at room temperature. RNAs were purified from the lysates using the PureLink RNA Mini Kit (Thermo Fisher). Purified RNA samples were reverse-transcribed using RNA to cDNA EcoDry Premixes (Takara Bio Inc., Kyoto, Japan). The total reaction volume was adjusted to 20 µl with RNase-free water after mixing with 100 ng cDNA, primer sets, SYBR Premix Ex Taq (Takara Bio Inc.), and ROX reference dye (Takara Bio Inc.). Amplification was performed with StepOne Plus (Applied Biosystems, Waltham, MA, USA) under the following cycling conditions: 95 °C for 30 s, followed by 40 cycles of 95 °C for 5 s, and 60 °C for 30 s. Analyses were performed in five independent experiments for each cDNA. The relative mRNA expression levels were calculated using the comparative threshold cycle (Ct) method with *GAPDH* as a control, as follows: $\Delta Ct = Ct$ (GAPDH) − Ct (target gene). The fold-change in gene expression normalized to GAPDH and relative to the control sample was calculated as $2^{-\Delta\Delta Ct}$. Primers for qRT-PCR are as follows: *ATGL* forward: 5′-ACC AGC ATC CAG TTC AAC CT-3′, reverse: 5′-ATC CCT GCT TGC ACA TCT CT-3′, *MGL* forward: 5′-GGT CAA TGC AGA CGG ACA GT-3′, reverse: 5′-ATG GAG TGG CCC AGG AGG AA-3′, *GAPDH* forward: 5′-CAT CAC TGC CAC CCA GAA GAC TG-3′, reverse: 5′-ATG CCA GTG AGC TTC CCG TTC AG-3′.

**Fly studies.** Fly stocks: Flies were maintained on instant fly food purchased from Carolina (Burlington, NC, #173200) at 25 °C and 60% humidity with a 12 h/12 h light/dark cycle. Mutant fly strains *bmm*$^1$ and *UAS-bmm:EGFP* were provided by Dr. Ronald P. Kühnlein[17]. *NP1-GAL4* (also known as *Myo1A-GAL4*, DGRC112001) was obtained from the Kyoto Stock Center. The *dGrasp*$^{302}$ (BL65258), *UAS-dGrasp-GFP* (BL8507), *UAS-mCD8::mRFP* (BL27398), and *UAS-dGrasp-RNAi* (BL34082) strains were obtained from the Bloomington Stock Center. All the mutant lines and transgenic lines were back-crossed for five generations to the *w*$^{1118}$ control genotype. For clarity, the *w*$^{1118}$ line is referred to as control throughout the manuscript.

Fly intestine staining: Flies used for dissection were 7- to 14-day-old females. The guts, from foregut to rectum/anus, were dissected from the flies in 1× PBS. Isolated guts were fixed with 4% paraformaldehyde in 0.1% Triton X-100 PBS (PBST) for 30 min, and washed three times in 1× PBS for 15 min each. Fixed samples were incubated for 30 min with Nile Red solution (N3013, Sigma-Aldrich, 1 mg ml$^{-1}$ in acetone) for lipid staining, and TO-PRO-3 solution (T3605, Invitrogen, 1 mM ml$^{-1}$ in DMSO) for nuclear staining, each at a dilution of 1:1000 in 0.1% PBST. The samples were washed three times in 1× PBS for 15 min each and then mounted on a glass slide with Vectashield mounting medium (H-1000 Vector Laboratories). For enterocyte labeling in *NP1>mCD8::RFP* flies, fixed guts were incubated with blocking solution containing 5% normal goat serum in 0.1% PBST. Next, the samples were incubated with rabbit DsRed antibodies (1:200, 632496, Clontech) in PBST overnight at 4 °C, and then with Alexa 568 anti-rabbit (1:400, A11011, Molecular Probes) for 1 h at room temperature after three washes with PBST. For lipid staining in these flies, the samples were incubated with BODIPY solution (D3922, Thermo Fisher Scientific, 1 mg ml$^{-1}$) at a dilution of 1:1000 in 0.1% PBST for 30 min, and washed twice with PBST. Stained guts were mounted and visualized using a LSM 700 Zeiss confocal microscope (Jena, Germany).

TG measurement: The Triglyceride Colorimetric Assay kit (10010303, Cayman Chemical) and BCA Protein Assay kit (23227, Thermo Fisher Scientific) were used for measuring triglyceride levels and quantifying proteins, respectively. The heads of five female flies were removed, and the rest of the bodies were placed in 1.5-ml tubes containing 100 µl diluted NP40 substitute assay reagent (supplied with the kit). The samples were homogenized using a pestle, incubated at 70 °C for 5 min, and then centrifuged for 10 min at 4 °C. To quantify TG content, 10 µl of each sample was transferred to a 96-well plate and 150 µl of diluted enzyme mixture solution (supplied with the kit) was added to each well. The plate was then incubated for 15 min at room temperature, and the sample absorbances were measured at a wavelength of 540 nm. To normalize lipid levels, protein levels were measured from 10 µl of supernatant from each sample using the BCA assay kit, as follows: The BCA solution was prepared by mixing solution A and solution B in a ratio of 50:1. From each sample, 10 µl of supernatant were mixed with 200 µl of the BCA solution, placed in a 96-well plate, and incubated at 37 °C for 30 min. The absorbances were then measured at a wavelength of 562 nm to determine protein levels. Finally, the normalized lipid level for each fly genotype was calculated by dividing the OD value at 540 nm by the OD value at 562 nm.

**Statistical analysis**. All experiments were performed at least three times independently for each condition. The results of multiple experiments are presented as the mean ± SEM. Statistical analyses were performed using Student's *t*-tests or analysis of variance (ANOVA) followed by Tukey's multiple comparison tests as appropriate; $p < 0.05$ was considered statistically significant. Calculations were performed using GraphPad Prism5 (GraphPad Software, Inc., La Jolla, CA, USA).

**Reporting summary**. Further information on research design is available in the Nature Research Reporting Summary linked to this article.

## Data availability

The source data underlying Figs. 1–10, Supplementary Figs. 1–19 and Supplementary Tables 2 and 3 are provided as a Source Data file. All data are available from the corresponding author upon reasonable request.

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

## Acknowledgements

We thank Drs. Catherine Rabouille, Hubrecht Institute, The Netherlands, and Sahng Wook Park, Yonsei University, Korea for valuable discussions and suggestions. We also thank Dong Soo Chang for editorial assistance and the Yonsei-Carl Zeiss Advanced Imaging Center for technical assistance. This work was supported by funding from NRF-2013R1A3A2042197 (to M.G.L.), and 2014M3A9D5A01073886 (to C.S.C.) from the National Research Foundation, the Ministry of Science, ICT & Future Planning, Republic of Korea.

## Author contributions

J.K. and M.G.L. conceived of the concept, designed the experiments, and wrote the paper; J.K., S.H.N., D.G.J., S.-Y.P., D.M., Hyunki K., Hoguen K., S.A., S.S., C.S.C., Hail K., J.W.K., and H.Y.G. performed molecular, histological, and metabolic studies in mice and

cultured cells; Hyeyon K. and S.J.M. performed fly works; H.-S.K. performed electron-microscopic examinations.

## Competing interests

The authors declare no competing interests.
