## [Peer Review File · Nature Communications]

Reviewers' comments:

Reviewer #1 (Remarks to the Author):

The manuscript from Kim et al., characterizes the role of the Golgi protein Grasp55 on lipid absorption and LD biology. Specifically, the authors show that ablation of Grasp55 impairs chylomicron processing in the gut to robustly reduce tissue lipid stores, effects that are also observed in flies. Moreover, they go on to show that impaired targeting of ATGL to lipid droplets in enterocytes appears to be an underlying mechanism of the reduced lipid packaging. Overall, these are exciting findings that provide new insights into the regulation of lipid absorption and processing as well as providing new information on ATGL trafficking and LD biology. Overall, I found this to be a very comprehensive and thorough manuscript that is well-written. The data analysis is done well and the level of detail given the methods is admirable. With that said there are two key points listed below that I believe need to be clarified for accurate interpretation of the data.

Figure 2C shows EM pics of enterocyte LDs. Interestingly, the supersized LDs in the Grasp55 KO mice appear to be enclosed in some sort of vesicle/compartiment. It is hard to tell from the EM images, but could these be autophagosomes or another form of an autophagy vesicle? Autophagy and lysosomal lipid degradation are becoming more appreciated for their roles in LD turnover in the gut (as well as other tissues). If these are autophagosomes or swollen lysosomes, then perhaps this would explain why ATGL can't access the LDs and subsequently gets degraded. Related, the Golgi also influences autophagy further suggesting that autophagy could be linked to the observed outcomes.

While it is clear that lipid accumulation is reduced in tissues, it is unlikely that a reduction in lipid content can account for the very large decrease observed in tissue weights and animal size. This brings up the concern that Grasp55 may be affecting development. Has this been ruled out...or at least considered?

Reviewer #2 (Remarks to the Author):

NCOMMS-18-28394-T

Grasp55^{-/-} mice display impaired fat absorption and resistance to high-fat diet-induced obesity
Jiyoon Kim, Hyeyon Kim, Shin Hye Roh, Shi-Young Park, Dong-Kook Min, Hyunki Kim, Hee-Seok Kweon, Hoguen Kim, Sowon Aum, Sookyung Seo, Cheol Soo Choi, Hail Kim, Jae-woo Kim, Seok Jun Moon, Heon Yung Gee, and Min Goo Lee

In this study the authors knocked out the Golgi stacking protein GRASP55 in mice and fly and found that GRASP55 knockout impairs fat absorption, chylomicron assembly and secretion, and increased the size of lipid droplets (LD) in intestinal epithelial cells. They reasoned this as defective targeting of some LD-associated lipases, such as ATGL and MGL, from the Golgi to LDs. These effects could be rescued by exogenously expressed GRASP55. The topic is in general interesting and some of the findings are novel. The main concern is that there is a lack of mechanism on the role of GRASP55 in the related processes. There are also a few major concerns on the experimental results.

1. The authors provided results that GRASP55 knockout impaired intestinal fat absorption, reduced chylomicron secretion, increased the size but reduced the number of LD. The cause and effect relationship between these changes was not well described. More importantly, it is not clear how GRASP55 coordinates these processes. A diagram or model on GRASP55 functions in these processes may help the reader to understand the mechanisms.

2. Similarly, how does GRASP55 increase the size but reduce the number of LD?

3. The authors claimed that a main defect in the GRASP55 KO tissue or cells is defective targeting of ATGL and MGL from the Golgi to LDs. The evidence for this conclusion is weak. Based on Figure 4e,f, the level of ATGL is significantly reduced in GRASP55 KO tissues; while in Figure 5a, the LD localization of ATGL is not convincing. This raises a possibility that a low protein level of ATGL instead of mistargeting may be the main reason of reduced chylomicron secretion and increased LD size.

4. Related to above, there is limited information on how GRASP55 modulates ATGL trafficking. The authors showed that GRASP55-ATGL interaction was increased under lipid-rich conditions (Figure 5f,g). Does this mean the interaction occurs on LD? If so the authors need to show GRASP55 LD localization. To figure out how G55 regulates ATGL trafficking from the Golgi to the LD, the authors may identify the ATGL-GRASP55 binding site followed by mutagenesis and addback experiments.

5. APOB is an important protein, but the results in Figure 4a,b and Supplementary Figure 8 are of low quality (so are some other proteins). The results are also inconsistent with Figure S5. APOB level, localization and secretion under different conditions should be thoroughly analyzed.

6. Figure 2e, the enlarged EM images showed some double membrane structures. GRASP55 has recently been shown to play a role in autophagy; GRASP55 depletion increases the number of autophagosomes. In addition, autophagy has been shown to be involved in LD turnover. Are the authors sure that these are indeed chylomicrons? The authors need to check whether reduced autophagy flux contributes to the accumulation of enlarged LD.

7. Since GRASP55 has been shown to be an essential Golgi structure protein, a detailed analysis of the Golgi structure and function with quantitation is needed. The authors do not formally quantify Golgi stacking in GRASP55 $-/-$ mice, yet state that "...depletion of GRASP55 alone among several Golgi cisternal adhesive proteins did not disrupt the Golgi stacking structure of mouse enterocytes." Furthermore, the authors fail to provide evidence that GRASP deletion in mice does not change the steady-state levels and localization of other Golgi cisternal adhesive proteins, such as GM130, Golgin-45, GRASP65, among many others.

8. In general the authors should indicate the number of mice used in each experiment and declare how many times the experiment was completed. The same principle applies to in vitro experiments. Particularly, if S.E.M. is used. Otherwise, it is unclear whether the mean actually represents the mean of multiple experiments where the mean was calculated or simply represents the mean from a single experiment.

Minor points:

1. Figure 1, the body weight of WT and GRASP55 KO mice are 35 g and 20 g, respectively. Can the fat amount count for the main difference?

2. Figure 2C, it is assumed that the upper panel is WT and the lower panel is KO. But this is not labeled on the figure.

3. Figure 2a,b, a quantitative assessment would improve the strength of the author's argument that GRASP deletion caused increased lipid storage of intestinal epithelial cells.

4. Figure 3e,f, addback experiments would strengthen the author's argument that GRASP55 regulates transepithelial lipid absorption. In addition, GRASP55 knockdown may affect the cell shape, motility and differentiation. Has this been taken into account?

5. Figure 5, addback experiments are essential for validating the author's argument that GRASP55 regulates LD-targeting of ATGL in Caco-2 and HeLa cells.

6. Figure 6, what's the relative level of exogenous GRASP55 vs. endogenous GRASP55 in WT tissues? Is the exogenous GRASP55 correctly localized?

7. Figure 7A, the authors should deploy a similar experimental strategy to determine if overexpression of mammalian ATGL rescues impaired fat absorption and resistance to high fat diet in GRASP55 ^{-/-} mice.

Reviewer #3 (Remarks to the Author):

In their study Kim and colleagues address an evolutionarily conserved role of the Golgi reassembly stacking protein of 55 kDa (GRASP55) in storage lipid homeostasis. The authors use a broad spectrum of mouse and fly knockout approaches along with tissue culture analyses to reveal that GRASP55 affects intestinal lipid absorption by Golgi-mediated lipid droplet targeting of lipases, in particular of ATGL. The authors show that GRASP55 knockout mice suffer from intestinal lipid accumulation and reduced chylomicron secretion. Moreover, they claim that these intestinal defects are causative for the hypotrophy of adipose tissue and hepatocytes and give rise to reduced body fat mass and growth retardation.

The study addresses an interesting, important, and timely question. The wealth of data and approaches is impressive and with few exceptions (see below) the experimental design is adequate, the results are convincing, and the conclusions are well-supported.

The cardinal weakness of the study, however, is the lack of tissue/organ-specific mutant analyses (or tissue/organ-specific rescue experiments) in mice. The extent of the contribution of the lack of GRASP55 in intestinal epithelial cells to the organismal phenotypes of the mutant animals remains questionable.

Points of concern:

To figure 1:

The authors point out that there is no significant weight gain of Grasp55 knockout mice on ND after 12 weeks of age. What is the food intake and the energy expenditure of these mutant mice compared to controls? Weight and growth differences between knockouts and control mice are apparent already few days after birth. Also, size/weight reductions appear not to be restricted to the adipose tissues but seem to also affect lean mass. How do these reductions scale to the overall growth retardation (illustrated in figure 1d)?

Looking at figure 1e, makes it hard to follow the argument of the authors that Grasp55 knockout mice are resistant to HFD-induced weight gain. The comparison to Grasp55 on ND is not informative.

To figure S3:

Is it correct that the fasting plasma leptin normalized to fat mass is increased in Grasp55 knockout mutants (compared to controls) as the fat mass is reduced? If so, what is the physiological interpretation of the authors?

To figures S3 and S4: It would be more informative for the readers to see key findings of those suppl. figures moved to main figure 1.

To figure 3:

I wonder if the defect in lipid absorption is specific to TG. The authors should also include studies using cholesterol to address this issue. Moreover, biochemical assessment of intestinal TG, cholesterol and cholesteryl ester content should be included. Chylomicron secretion rates should be measured more directly using the established tyloxapol-based methods.

3a: There is a discrepancy between main text and figure caption concerning the age of the mice subjected to OFTT.

3e-g: The Caco-2 experiments are a powerful approach to disentangle the dynamics of the cellular processes underlying the mouse mutant intestinal phenotypes. Yet, the interpretation needs some explanation in particular concerning the intracellular TG handling. Does reduced transit from A to B correlate with increased LD formation in the Caco-2 cells as seen in the intestinal epithelium? Similar experiments illustrated in figure 5a seem to argue against LD accumulation in fed Caco-2 cells subjected to Grasp55 knockdown. With other words is there evidence for a retention of labeled FA in the TG pool? It would be also interesting to know if the FA exits the cell at the basolateral side as TG or FA.

To figure 4:

4a,b: The subcellular fractionation experiments suggest the presence of lipases in the Golgi fraction. How do authors exclude that this is due to a cross-contamination with cytosol or lipid droplets? The authors need to include bona fide LD and cytosolic markers in the Golgi/ERGIC/ER fractionation experiment.

Also, the authors report that experiments shown in this panels were independently repeated three times. Do the data shown in 4a,b and S8 represent independent experiments? If not, showing the same data twice needs to be described.

4c: Comparing the HSL and pHSL data between 4c with the corresponding quantifications in S11 raises doubts concerning the representative nature of the blot shown in 4c.

4e: ATGL immunoreactivity is indeed reduced in the LD and cytosolic fraction of Grasp55^{-/-} samples but unchanged in total membranes, which include the Golgi. Do the authors have an explanation for this discrepancy? Moreover, why is there β -actin detectable in the LD fraction?

To figure 5:

The immunostaining of ATGL in Caco-2 cells is of poor quality. ATGL immunoreactivity is detected all over the cell (even in the nucleus) but is hardly enriched specifically at the LDs. It is questionable, whether this staining permits a reliable readout of LD targeting. The authors need to use cell fractionation as a complementary method to validate this finding.

5f,g: A reciprocal IP using anti-ATGL would substantiate the surprising and interesting finding that Grasp55 and ATGL physically interact.

To line 194/195: What is the evidence from this work that specifically the "non-LD-associated ATGL" is subject to proteasomal degradation (and not also the LD pool)?

To figure 6:

Ubiquitous over-expression of Grasp55 provides solid genetic evidence that the observed metabolic effects are indeed due to the knockout of Grasp55. Also, there is convincing evidence that lack of Grasp55-dependent lipid absorption, involving the proposed mechanism of lipase localization/protein turnover, contributes to the mutant phenotype. Yet in the absence of tissue-specific rescue experiments the impact of this mechanism on the mutant phenotype remains elusive.

How does the protein expression level from the myc-tagged Grasp55 transgene relate to the endogenous expression? Figure 6h,i seems to indicate that intestinal ATGL abundance appears to be higher in the rescued mutants compared to controls.

To figure 7:

The *Drosophila* data represent a meaningful contribution to the study. However, important details are not described and there is no evidence for an epistatic relation between dGrasp and bmm in the control of midgut lipid homeostasis. In particular, the following questions need to be addressed: Where along the midgut does lipid accumulate in the dGrasp mutants? Do midgut enterocytes or other cell types accumulate lipids in this mutant condition? Is Bmm expressed in these cells? How does the bmm mutant phenotype look like in these cells/this region of the midgut? If bmm were downstream of dGrasp, (i) bmm mutants are expected to show a dGrasp-like ectopic lipid accumulation phenotype, and (ii) the double mutant could be reverted by providing bmm but not by providing dGrasp expression under the control of NP1.

Rajan et al. (ref. 18; fig. 2B) report reduced body fat content in response to fat body (=adipose tissue+liver)-specific dGrasp55 in vivo knockdown, which – at least for flies - challenges the model of the authors that adipose tissue hypotrophy in Grasp55 mutant mice is secondary to the lipid absorption defect (and not a consequence of cell autonomous effects in this tissue). These results also need to be discussed in the light of the reversion of the body fat content of dGrasp mutant flies subject to NP1-driven intestinal dGrasp expression.

General remarks:

Line 36: "gene silencing" is an inappropriate description for a deletion mutant.

The authors are advised to carefully reinspect their claims concerning the cell/tissue-specificity of their experiments. E.g. they refer to enterocytes (line 179) but use complete intestine samples.

Line 261: Beller et al. PLoS Biology 2008 already related ATGL degradation to failure in COPI trafficking.

Attention needs: figure 5f „IP:Ani-Myc“, "evolutionally" in line 231, "upstream activation signal" line 821.

TG content measurements on flies appears to be missing from the Methods section.

The authors are advised to avoid calling w[1118] mutant flies wild-types. Designation as controls is correct and informative.

The manuscript would benefit from another round of English proofreading to avoid e.g. „... except for defected spermatogenesis ...“ (line 53).

Point-by-point Response to Reviewers' Comments

Dear Editor and Reviewers,

We appreciate the time and effort you have put into reviewing our manuscript. Our point-by-point responses to the comments are presented below. The comments from **Reviewers** are provided in **bold font** with numbering, while our responses are below in regular font. In the revised manuscript, all changes are indicated in red font.

Reviewer #1

(Remarks to the Author)

The manuscript from Kim et al., characterize the role of the Golgi protein Grasp55 on lipid absorption and LD biology. Specifically, the authors show that ablation of Grasp55 impairs chylomicron processing in the gut to robustly reduce tissue lipid stores, effects that are also observed in flies. Moreover, they go on to show that impaired targeting of ATGL to lipid droplets in enterocytes appears to be an underlying mechanism of the reduced lipid packaging. Overall, these are exciting findings that provide new insights into the regulation of lipid absorption and processing as well as providing new information on ATGL trafficking and LD biology. Overall, I found this to be a very comprehensive and thorough manuscript that is well-written. The data analysis is done well and the level of detail given the methods is admirable. With that said there are two key points listed below that I believe need to be clarified for accurate interpretation of the data.

Comment 1.

Figure 2C shows EM pics of enterocyte LDs. Interestingly, the supersized LDs in the Grasp55 KO mice appear to be enclosed in some sort of vesicle/compartment. It is hard to tell from the EM images, but could these be autophagosomes or another form of an autophagy vesicle? Autophagy and lysosomal lipid degradation are becoming more appreciated for their roles in LD turnover in the gut (as well as other tissues). If these are autophagosomes

or swollen lysosomes, then perhaps this would explain why ATGL can't access the LDs and subsequently gets degraded. Related, the Golgi also influences autophagy further suggesting that autophagy could be linked to the observed outcomes.

>>Response:

1) The double membrane-like structures in the EM images in previous figures are probably due to the shrinkage of supersized lipid droplets; this is one of the common artifacts of EM imaging of lipid-rich structures, and occurs during the chemical fixation or acquisition of EM images using high-intensity electron beams. We re-conducted the EM analyses with great care to minimize this “shrinkage artifact” of supersized LDs. New Fig. 3e (revision of previous Fig. 2c) mostly shows the single membrane structure of supersized LDs. For clarification, we also provide an enlarged EM image of a supersized LD in the figure for Reviewer 1.

Figure for Reviewer 1. **a**, Full-size image of Fig. 3e. **b**, Enlarged image of inset in **a**. The supersized LDs show a single membrane structure.

2) To address the suggestion by the Reviewer that deletion of GRASP55 may affect autophagy, we examined the conversion of microtubule-associated protein light chain 3 (LC3) into its phosphatidylethanolamine-conjugated form (LC3-II), a parameter of autophagic flux, in the jejunums of mice fasted (16 h) or 2 h after olive oil feeding. As shown in Supplementary Fig. 18, *Grasp55* deletion did not significantly affect autophagic flux in the mouse jejunum, suggesting that altered autophagy is not responsible for the formation of supersized LDs in the *Grasp55*-null mice. Relevant descriptions have also been added in the text (lines 224-230).

Comment 2.

While it is clear that lipid accumulation is reduced in tissues, it is unlikely that a reduction in lipid content can account for the very large decrease observed in tissue weights and animal size. This brings up the concern that Grasp55 may be affecting development. Has this been ruled out...or at least considered?

>>Response:

We thank the Reviewer for pointing out this important issue. At the age of 30 weeks, *Grasp55*-null mice showed a significant body weight reduction ($-41.4 \pm 2.6\%$); this was fully restored upon high-fat diet feeding (Fig. 1e), suggesting that decreased lipid uptake and the associated nutritional imbalance are the major problems causing growth retardation. However, as suggested by the Reviewer, other factors may also contribute to growth retardation, particularly a reduction in lean body mass. A potential factor would be a decrease in plasma insulin level (Fig. 2e). Although it did not cause differences in fasting plasma glucose levels (Fig. 2f), a sustained reduction in plasma insulin levels may contribute to growth retardation, in particular reduced lean body mass, of *Grasp55*^{-/-} mice. This is now discussed in the revised text (lines 374-377).

Reviewer #2

In this study the authors knocked out the Golgi stacking protein GRASP55 in mice and fly and found that GRASP55 knockout impairs fat absorption, chylomicron assembly and secretion, and increased the size of lipid droplets (LD) in intestinal epithelial cells. They reasoned this as defective targeting of some LD-associated lipases, such as ATGL and MGL, from the Golgi to LDs. These effects could be rescued by exogenously expressed GRASP55. The topic is in general interesting and some of the findings are novel. The main concern is that there is a lack of mechanism on the role of GRASP55 in the related processes. There are also a few major concerns on the experimental results.

Comment 1.

The authors provided results that GRASP55 knockout impaired intestinal fat absorption, reduced chylomicron secretion, increased the size but reduced the number of LD. The cause and effect relationship between these changes was not well described. More importantly, it not clear how GRASP55 coordinates

these processes. A diagram or model on GRASP55 functions in these processes may help the reader to understand the mechanisms.

>>Response:

We thank the Reviewer for your suggestion. We now provide a schematic diagram that summarizes the major findings of present study, in Supplementary Fig. 23. Briefly, some LD-associated proteins, such as ATGL and MGL, reach their final destination of LDs via a Golgi- and GRASP55-dependent route. Defects in this process induce supersized LDs and reduced lipidation of chylomicrons in enterocytes upon exogenous lipid challenge, which eventually results in reduced fat absorption (lines 379-382).

Comment 2.

Similarly, how does GRASP55 increases the size but reduces the number of LD?

>>Response:

LD sizes differ considerably between cells and vary across different emulsion scales. There are three major processes that can increase the size of LDs: fusion (coalescence), ripening, and growth by new TG synthesis (PMID: 24220094). In the *Grasp55*^{-/-} mouse intestine upon exogenous lipid challenge, TGs would be continuously accumulated in LDs due to lack of ATGL and MGL, which enriches LDs in the confined apical area of enterocytes. This causes two LDs to come into close proximity and increases the possibility of fusion events, which initially increases the size of certain LDs. A second mechanism of LD destabilization is ripening, in which small droplets of an emulsion disappear as bigger ones grow. In this case, the bigger LDs absorb surrounding small LDs, which eventually produces reduced numbers of supersized LDs. Although the classical Oswald ripening is difficult occur *in vivo*, various mechanisms, such as detergents and specific proteins (e.g., Fsp27/CIDEA) can facilitate the ripening process (PMID: 24220094). A more sophisticated biophysical experiment is required to fully resolve this question.

We would like to mention that cells tend to have a small number of large-sized LDs when the intracellular lipid content is increased. For example, adipocytes often contain a single LD of tens or hundreds of microns (PMID: 24220094). Interestingly, similar findings were observed in liver tissues in the present study. In this case, *Grasp55*^{+/+} hepatocytes, which contain higher amounts of TGs, show large-sized but reduced numbers of LDs (Supplementary Fig. 19e, f). Therefore, increased TG levels, rather than the GRASP55 expression status, appear to be primarily responsible for the increased LD size but reduced LD number in *Grasp55*^{-/-} enterocytes.

Comment 3.

The authors claimed that a main defect in the GRASP55 KO tissue or cells is defective targeting of ATGL and MGL from the Golgi to LDs. The evidence for this conclusion is weak. Based on Figure 4e,f, the level of ATGL is significantly reduced in GRASP55 KO tissues; while in Figure 5a, the LD localization of ATGL is not convincing. This raises a possibility that a low protein level of ATGL instead of mistargeting may be the main reason of reduced chylomicron secretion and increased LD size.

>>Response:

We agree that low protein levels of ATGL may contribute to reduced chylomicron secretion and increased LD size. The most critical point is the absence of ATGL on LDs. Both the mistargeting of ATGL and the reduction in total ATGL protein levels contribute to the phenotype of *Grasp55*^{-/-} mice caused by the absence of ATGL on LDs. We believe that the mistargeting and low protein levels of ATGL are not independent events, but are closely related. Our results showed that inhibition of Golgi traveling of ATGL by GRASP55 deletion (Fig. 5a) inhibited the LD localization of ATGL (Fig. 5d) and caused proteasome-mediated ATGL degradation (Supplementary Fig. 15). Similarly, it has been previously shown that inhibition of COPI-mediated ATGL transport by GBF1 depletion inhibited the LD localization of ATGL and evoked proteasome-mediated ATGL degradation (Fig. 3 in Ref #27). Therefore, the mislocalization of ATGL is primarily responsible for ATGL degradation and its low protein levels in cells. For this reason, we focused on the mistargeting of ATGL as the main reason for reduced chylomicron secretion and increased LD size. However, as indicated by the Reviewer, a low protein level of ATGL may additionally aggravate the phenotype of GRASP55 deficiency, and the text has been revised accordingly (lines 382-383).

Comment 4.

Related to above, there is limited information on how GRASP55 modulates ATGL trafficking, The authors showed that GRASP55-ATGL interaction was increased under lipid-rich conditions (Figure 5f,g). Does this mean the interaction occurs on LD? If so the authors need to show GRASP55 LD localization. To figure out how G55 regulates ATGL trafficking from the Golgi to the LD, the authors may identify the ATGL-GRASP55 binding site followed by mutagenesis and addback experiments.

>>Response:

We thank the Reviewer for these important experimental suggestions.

1) Immunostaining results showed that GRASP55 did not localize into the LDs even under lipid-rich conditions (Supplementary Fig. 16). This suggests that an interaction between GRASP55 and ATGL does not occur, at least at the LD. Considering that GRASP55 and ATGL colocalize at the Golgi (Fig. 5a), the GRASP55-ATGL interaction appears to occur at the Golgi complex (lines 211-214).

2) The main reason for the increased GRASP55-ATGL interaction under lipid-rich conditions is an increase in ATGL protein levels, not an increase in binding affinity between GRASP55 and ATGL, because the increase in GRASP55-ATGL binding was proportional to cellular ATGL levels and treatment with oleic acid (lipid-rich condition) did not affect the level of 'IP fold change/ATGL in cell lysate' (new Fig. 7b, d; previous Fig. 5g). The relevant text has been revised (lines 209-211).

3) As requested, we performed a pull-down assay using protein fragments of GRASP55 and ATGL to elucidate the GRASP55-ATGL interaction (Fig. 7e, f). The results indicated that the GRASP55-PDZ1 domain specifically interacts with the ATGL-patatin domain. Previous studies have shown that the patatin domain interacts with many other regulatory proteins of ATGL, such as CGI-58, HILPDA, and G0S2 (Ref. #28). Relevant passages have now been added in the text (lines 214-217).

4) As suggested by the Reviewer, we performed a series of add-back experiments that would increase the strength of the present study. These are now described in the text (pages 7-9).

i) Fig. 4e–g: GRASP55 depletion (*shGRASP55*)-induced transepithelial lipid absorption defects were rescued upon supplementation with exogenous GRASP55 (GRASP55-Myc) in Caco-2 cells (lines 157-158).

ii) Fig. 6c and Supplementary Fig. 14: The LD targeting failure of ATGL upon GRASP55 depletion was rescued upon supplementation with exogenous GRASP55 in Caco-2 cells (Fig. 6c) and HeLa cells (Supplementary Fig. 14) (lines 202-204).

iii) Fig. 6d, e and Supplementary Fig. 15: Reduction in ATGL levels and protein stability by GRASP55 depletion was reversed upon supplementation with exogenous GRASP55 in Caco-2 cells (lines 202-204).

iv) Fig. 6f, g: Exogenous ATGL (ATGL-Myc) supplementation rescued the defects in transepithelial lipid absorption in GRASP55-depleted Caco-2 cells (lines 206-208).

Comment 5.

APOB is an important protein, but the results in Figure 4a,b and Supplementary Figure 8 are of low quality (so are some other proteins). The results are also inconsistent with Figure S5. APOB level, localization and

secretion under different conditions should be thoroughly analyzed.

>>Response:

1) As requested, we have examined the changes in APOB protein levels under fasting and lipid-rich (olive oil feeding) conditions. The results showed that the plasma APOB levels of *Grasp55*-null mice were reduced under both fasting and lipid-rich conditions (Supplementary Fig. 3a, b). However, GRASP55 deficiency did not affect APOB protein levels in the intestines (Supplementary Fig. 3c, d). Relevant comments have now been added in the text (lines 138-141).

2) We have also examined APOB localization in the intestines under fasting and lipid-rich conditions. APOB proteins were concentrated in the subapical area in both *Grasp55*^{+/+} and *Grasp55*^{-/-} enterocytes under fasting conditions. Most APOB proteins were translocated to the basolateral area after olive oil feeding in both *Grasp55*^{+/+} and *Grasp55*^{-/-} enterocytes. Interestingly, APOBs in *Grasp55*^{+/+} enterocytes were more highly concentrated near the basolateral membrane after olive oil feeding (Supplementary Fig. 3e).

Comment 6.

Figure 2e, the enlarged EM images showed some double membrane structures. GRASP55 has recently been shown to play a role in autophagy; GRASP55 depletion increases the number of autophagosomes. In addition, autophagy has been shown to be involved in LD turnover. Are the authors sure that these are indeed chylomicrons? The authors need to check whether reduced autophagy flux contributes to the accumulation of enlarged LD.

>>Response:

Please see the response to Comment 1 of Reviewer 1. Briefly, the double membrane-like structures in the previous EM image of LDs appear to be an artifact of EM imaging of lipid-rich structures. The Fig. 3e (previous Fig. 2d) mostly shows the single-membrane structure of supersized LDs (see also Figure for Reviewer 1). As shown in Supplementary Fig 18, GRASP55 deficiency did not significantly affect the autophagic flux in the mouse jejunum, suggesting that altered autophagy is not responsible for the formation of supersized LDs in the *Grasp55*-null mice. Relevant descriptions have now been added in the text (lines 224-230).

Comment 7.

Since GRASP55 has been shown to be an essential Golgi structure protein, a detailed analysis of the Golgi structure and function with quantitation is needed. The authors do not formally quantify Golgi stacking in GRASP55 -/-

mice, yet state that “..depletion of GRASP55 alone among several Golgi cisternal adhesive proteins did not disrupt the Golgi stacking structure of mouse enterocytes.” Furthermore, the authors fail to provide evidence that GRASP deletion in mice does not change the steady-state levels and localization of other Golgi cisternal adhesive proteins, such as GM130, Golgin-45, GRASP65, among many others.

>>Response:

As suggested, we have examined the protein levels and localization of several Golgi cisternal adhesive proteins. The results indicate that the deficiency of GRASP55 did not affect the protein levels of GM130, Golgin97, GRASP65, and Golgin45 (new Supplementary Fig. 8a, b). In addition, GRASP55 deficiency did not alter the Golgi localization (apical side of perinuclear regions in the jejunal epithelia) of GM130, Golgin97, GRASP65, and Golgin45 (Supplementary Fig. 8c). Relevant descriptions have now been added in the text (lines 168-171).

Comment 8.

In general the authors should indicate the number of mice used in each experiment and declare how many times the experiment was completed. The same principle applies to in vitro experiments. Particularly, if S.E.M. is used. Otherwise, it is unclear whether the mean actually represents the mean of multiple experiments where the mean was calculated or simply represents the mean from a single experiment.

>>Response:

In all figures and supplementary figures, we have indicated the number of mice used in each experiment and number of experimental replicates. These are now marked in red in each figure legend.

Minor points:

1. Figure 1, the body weight of WT and GARS55 KO mice are 35 g and 20 g, respectively. Can the fat amount count for the main difference?

>>Response:

Please see the response to Comment 1 of Reviewer 1. Briefly, the body weight reduction in GRASP55 KO mice was restored by high-fat diet feeding (Fig. 1e), suggesting that decreased lipid uptake and the associated nutritional imbalance are the major problems causing the growth retardation. However, other factors, in particular a sustained reduction in plasma insulin levels, may also contribute to the

reduced lean body mass of *Grasp55*^{-/-} mice. This is now discussed in the revised text (lines 374-377).

2. Figure 2C, it is assumed that the upper panel is WT and the lower panel is KO. But this is not labeled on the figure.

>>Response:

We thank the Reviewer for pointing out this oversight. We have added the corresponding labels in the figure (new Fig. 3e in the revised manuscript).

3. Figure 2a,b, a quantitative assessment would improve the strength of the author's argument that GRASP deletion caused increased lipid storage of intestinal epithelial cells.

>>Response:

We thank the Reviewer for this constructive suggestion. We have added the results of quantitative analysis of Oil Red O absorbance measured at 510 nm (Fig. 3c). In addition, we added the results of TG measurements in the jejunums of *Grasp55*^{+/+} and *Grasp55*^{-/-} mice that were fasted for 16 h, or 4 h after olive oil bolus (Fig. 3d). The results consistently indicated that *Grasp55* deletion caused increased lipid storage in mouse intestinal epithelial cells (line 125).

4. Figure 3e,f, addback experiments would strengthen the author's argument that GRASP55 regulates transepithelial lipid absorption. In addition, GRASP55 knockdown may affect the cell shape, motility and differentiation. Has this been taken into account?

>>Response:

1) Please see the response to Major Comment 4. We performed the requested addback experiments (Fig. 3e–g in the previous manuscript) and the results are shown in the new Fig. 4e–g: GRASP55 depletion (*shGRASP55*)-induced transepithelial lipid absorption defects were rescued upon supplementation with exogenous GRASP55 (GRASP55-Myc) in Caco-2 cells.

2) As suggested, we have investigated cell shape and differentiation ability using confocal microscopy and by measuring the transendothelial electrical resistance (TEER), respectively, in differentiated Caco-2 monolayers. The results indicated that GRASP55 depletion did not noticeably affect cell shape and differentiation rate in Caco-2 cells. (Figures for Reviewer 2 and 3).

Figure for Reviewer 2. Cell shapes of differentiated Caco-2 cells. Control and GRASP55-depleted (*shGRASP55*) Caco-2 cells cultured on a permeable support for 10 days were stained with anti-ZO1 antibodies (tight junction, green) and rhodamine-phalloidin (F-actin, red). Nuclei were counterstained with 4',6-diamidino-2-phenylindole (blue). **a**, Representative XY-scan and Z-scan images of Caco-2 monolayers. **b**, The Z-distance of Caco-2 monolayers is summarized (n = 10). GRASP55 knockdown did not alter the vertical thickness of the Caco-2 monolayers. Scale bars: 20 μm. Data are shown as mean ± SEM. n.s.: not significant.

Figure for Reviewer 3. Measurements of transepithelial electrical resistance (TEER) in Caco-2 monolayers. Control and GRASP55-depleted (*shGRASP55*) Caco-2 cells were cultured on a permeable support for 10 days and TEER was measured using a protocol described previously (PMID: 25586998). GRASP55 depletion did not alter the development

of electrical resistance, suggesting that GRASP55 did not affect the differentiation of Caco-2 cells.

5. Figure 5, addback experiments are essential for validating the author's argument that GRASP55 regulates LD-targeting of ATGL in Caco-2 and HeLa cells.

>>Response:

Please see the response to Major Comment 4. We performed the requested addback experiments, and the results provided additional support that GRASP55 regulates LD-targeting of ATGL in Caco-2 and HeLa cells.

1) Fig. 6c (previous Fig. 5a) and Supplementary Fig. 14 (previous Supplementary Fig. 13): The LD targeting failure of ATGL upon GRASP55 depletion was rescued upon supplementation with exogenous GRASP55 in Caco-2 cells (Fig. 6c) and HeLa cells (Supplementary Fig. 14).

2) Fig. 6d, e (previous Fig. 5b, c) and Supplementary Fig. 15 (previous Fig. 5d, e): The reduction in ATGL levels and protein stability by GRASP55 depletion was reversed upon supplementation with exogenous GRASP55 in Caco-2 cells.

6. Figure 6, what's the relative level of exogenous GRASP55 vs. endogenous GRASP55 in WT tissues? Is the exogenous GRASP55 correctly localized?

>>Response:

1) We have included immunoblot images showing the differences in protein level between exogenous and endogenous GRASP55s in rescued mice (Fig. 8h). Quantitative analysis indicated that the protein expression of exogenous GRASP55 was approximately eight times higher than that of endogenous GRASP55 (lines 414-415).

2) We performed immunohistochemical analyses of exogenous GRASP55-Myc using anti-Myc antibodies in the mouse jejunum. Co-immunostaining with the Golgi marker protein GM130 revealed that transgenic GRASP55-Myc proteins were mostly localized on the Golgi in jejunal epithelial cells (Supplementary Fig. 20). Relevant comments have now been added in the text (lines 245-247).

7. Figure 7A, the authors should deploy a similar experimental strategy to determine if overexpression of mammalian ATGL rescues impaired fat absorption and resistance to high fat diet in GRASP55 -/- mice.

>>Response:

To resolve the question raised by the Reviewer, we measured transepithelial fat absorption in GRASP55-depleted Caco-2 cells with overexpression of mammalian

ATGL. It would take a very long time to perform these experiments in mice by newly generating Tg-ATGL mice and then mating them with *Grasp55*^{-/-} mice. As shown in Fig. 6f, g, overexpression of ATGL rescued the fat absorption defect of GRASP55-depleted Caco-2 cells, similar to that observed in flies (new Fig. 9c; previous Fig. 7a). Relevant comments have now been added in the text (lines 206-208).

Reviewer #3

In their study Kim and colleagues address an evolutionarily conserved role of the Golgi reassembly stacking protein of 55 kDa (GRASP55) in storage lipid homeostasis. The authors use an broad spectrum of mouse and fly knockout approaches along with tissue culture analyses to reveal that GRASP55 affects intestinal lipid absorption by Golgi-mediated lipid droplet targeting of lipases, in particular of ATGL. The authors show that GRASP55 knockout mice suffer from intestinal lipid accumulation and reduced chylomicron secretion. Moreover, they claim that these intestinal defects are causative for the hypotrophy of adipose tissue and hepatocytes and give rise to reduced body fat mass and growth retardation.

The study addresses an interesting, important, and timely question. The wealth of data and approaches is impressive and with few exceptions (see below) the experimental design is adequate, the results are convincing, and the conclusions are well-supported.

The cardinal weakness of the study, however, is the lack of tissue/organ-specific mutant analyses (or tissue/organ-specific rescue experiments) in mice. The extent of the contribution of the lack of GRASP55 in intestinal epithelial cells to the organismal phenotypes of the mutant animals remains questionable.

Points of concern:

To figure 1:

The authors point out that there is no significant weight gain of *Grasp55* knockout mice on ND after 12 weeks of age. What is the food intake and the energy expenditure of these mutant mice compared to controls?

>>Response:

We thank the Reviewer for bringing up these questions. We have performed a

metabolic cage study and examined energy intake (food intake) and energy expenditure under normal diet and high-fat diet conditions. GRASP55 deficiency did not alter total energy intake (Fig. 2a), but reduced total energy expenditure (Fig. 2b). Given the reduced energy expenditure and normal dietary food intake, the low body weight of *Grasp55^{-/-}* mice (Fig. 1e) implies that *Grasp55^{-/-}* mice may suffer from energy absorption. Interestingly, the reduced energy expenditure of *Grasp55^{-/-}* mice was significantly restored in mice fed the high-fat diet (Fig. 2b, inset). Relevant comments have now been added in the text (lines 101-105).

Weight and growth differences between knockouts and control mice are apparent already few days after birth. Also, size/weight reductions appear not to be restricted to the adipose tissues but seem to also effect lean mass. How does these reductions scale to the overall growth retardation (illustrated in figure 1d)?

>>Response:

1) Please see the response to Comment 1 of Reviewer 1. Briefly, the body weight reductions in *Grasp55*-null mice were restored upon high-fat diet feeding (Fig. 1e), suggesting that decreased lipid uptake and the associated nutritional imbalance are the major problems causing the growth retardation. However, other factors, in particular a sustained reduction in plasma insulin levels, may also contribute to the reduced lean body masses of *Grasp55^{-/-}* mice. This is now discussed in the revised text (lines 374-377)

2) The analyses in Supplementary Table 2 show that the absolute values of lean body mass were reduced in *Grasp55^{-/-}* mice. However, the ratio values (divided by the total body weight) of major organs, including skeletal muscle, were not significantly altered in *Grasp55^{-/-}* mice.

Looking at figure 1e, makes it hard to follow the argument of the authors that *Grasp55* knockout mice are resistant to HFD-induced weight gain. The comparison to *Grasp55* on ND is not informative.

>>Response:

To simplify and increase the readability of Fig. 1e, we have now divided the previous graph of body weight curves into two separate graphs of mice fed normal diet (ND) and high-fat diet (HFD) (new Fig. 1e). The new graphs more clearly show that the growth of *Grasp55^{-/-}* mice fed HFD is comparable to that of *Grasp55^{+/+}* mice fed ND, suggesting that *Grasp55^{-/-}* mice are resistant to HFD-induced weight gain.

To figure S3:

Is it correct that the fasting plasma leptin normalized to fat mass is increased in *Grasp55* knockout mutants (compared to controls) as the fat mass is reduced? If so, what is the physiological interpretation of the authors? To figures S3 and S4: It would be more informative for the readers to see key finding of those suppl. figures moved to main figure 1.

>>Response:

1) We thank the Reviewer for this insightful question. We found that the previous method of measuring plasma leptin levels (BioVision, #E4654-100) displayed poor sensitivity in detecting basal levels of plasma leptin. Therefore, we re-performed the whole experiments using a different assay kit that displayed better sensitivity (Cayman, #A05176). The new results indicated that fasting plasma leptin levels, as well as those after feeding, were reduced in *Grasp55*^{+/+} mice (new Fig. 2c). When fasting leptin levels were normalized to body fat mass, leptin levels were comparable between WT and KO mice (new Fig. 2d) (lines 109-112).

2) As suggested, we have now moved the data from the previous Supplementary Figs. 3 and 4 (plasma leptin and insulin levels) into a new main figure (new Fig. 2c-g).

To figure 3:

I wonder if the defect in lipid absorption is specific to TG. The authors should also include studies using cholesterol to address this issue. Moreover, biochemical assessment of intestinal TG, cholesterol and cholesteryl ester content should be included. Chylomicron secretion rates should be measured more directly using the established tyloxapol-based methods.

>>Response:

1) We now provide data showing the cholesterol content in plasma fractions 2 h after oral gavage with olive oil (new Supplementary Fig. 6a, b). The results indicate that the cholesterol content of the chylomicron/VLDL fraction was reduced in *Grasp55*^{-/-} mouse blood samples while those of IDL, LDL, and HDL were not altered, implying that cholesterol absorption via chylomicrons was also affected upon GRASP55 deficiency. Relevant comments have been added in the revised text (lines 152-155).

2) As requested, we measured the intestinal TG content, and the results are shown in Fig. 3d. Intestinal TG content was significantly increased in *Grasp55*^{-/-} mice after olive oil feeding, further supporting our finding that deletion of GRASP55 inhibited intestinal chylomicron secretion. We also measured intestinal cholesterol and cholesteryl ester content (new Supplementary Fig. 6c, d). Because olive oil does not contain cholesterol, the measurements were conducted following high-fat diet

feeding. The results indicated that deletion of GRASP55 did not significantly affect intestinal cholesterol and cholesteryl ester content.

3) We are grateful for the Reviewer's suggestion to experimentally determine the chylomicron secretion rate. As suggested, we re-performed the oral fat tolerance test with tyloxapol (Triton WR-1339); the results clearly indicate that GRASP55 depletion reduced the chylomicron secretion rate (Fig. 4a). Relevant comments have been added in the text (lines 142-145).

3a: There is a discrepancy between main text and figure caption concerning the age of the mice subjected to OFTT.

>>Response:

This has now been corrected. All mice are 12 weeks old (new Fig. 4 legend).

3e-g: The Caco-2 experiments are a powerful approach to disentangle the dynamics of the cellular processes underlying the mouse mutant intestinal phenotypes. Yet, the interpretation needs some explanation in particular concerning the intracellular TG handling. Does reduced transit from A to B correlate with increased LD formation in the Caco-2 cells as seen in the intestinal epithelium? Similar experiments illustrated in figure 5a seem to argue against LD accumulation in fed Caco-2 cells subjected to Grasp55 knockdown. With other words is there evidence for a retention of labeled FA in the TG pool? It would be also interesting to know if the FA exits the cell at the basolateral side as TG or FA.

>>Response:

We thank the Reviewer for these careful questions.

1) It has been shown that the majority of exogenous oleic acids are incorporated into TGs and secreted as TGs in Caco-2 cells (Ref. #52). Therefore, transepithelial lipid uptake experiments in Caco-2 monolayers with ¹⁴C-labeled oleic acid would be an appropriate model for intestinal lipid absorption and intracellular TG handling. This background information is now described in the figure legend (Fig. 4g).

2) Previously, we had performed immunostaining experiments 16 h after treatment with oleic acid (previous Fig. 5a). We have now carefully re-performed the experiments over a time course, and found that supersized LDs were present in GRASP55-depleted cells at earlier times (2–4 h after treatment with oleic acid) and disappeared with time. In the revised manuscript, we provide images taken at 4 h after oleic acid treatment, which clearly show supersized LDs in GRASP55-depleted Caco-2 cells (new Fig. 6c).

3) According to published literature (Ref. #52), it appears that Caco-2 cells secrete the majority of labeled FA toward the basolateral side as TGs.

To figure 4:

4a,b: The subcellular fractionation experiments suggest the presence of lipases in the Golgi fraction. How do authors exclude that this is due to a cross-contamination with cytosol or lipid droplets? The authors need to include bona fide LD and cytosolic markers in the Golgi/ERGIC/ER fractionation experiment.

>>Response:

We re-performed subcellular fractionation experiments of the mouse jejunum after oral gavage with olive oil to induce LDs. We separated the topmost lipid layer (LD fraction) and the subsequent fractions (cytosol, Golgi, ERGIC and ER fractions). Upon immunoblotting with organelle markers, including RAB18 (LD marker) and Aldolase A (cytosolic marker), we were able to confirm that ATGL and MGL are present in the Golgi fraction of *Grasp55^{+/+}* mouse jejunum, which is not contaminated with LDs and cytosolic proteins. Notably, ATGL and MGL were profoundly reduced in the Golgi fraction of *Grasp55^{-/-}* mice (new Fig. 5a).

Also, the authors report that experiments shown in this panels were independently repeated three times. Do the data shown in 4a,b and S8 represent independent experiments? If not, showing the same data twice needs to be described.

>>Response:

As described above, we re-performed the experiments shown in the new Fig. 5a (previous Fig. 4a,b) after olive oil gavage, and now provide the results independent of those in Supplementary Fig. 9 (previous Supplementary Fig. 8). We believe that this resolves the Reviewer's concern regarding showing the same data twice.

4c: Comparing the HSL and pHSL data between 4c with the corresponding quantifications in S11 raises doubts concerning the representative nature of the blot shown in 4c.

Response:

We have replaced the immunoblot images in previous Fig 4c with a different set of images (new Fig. 5b) that would represent the corresponding quantifications (new Supplementary Fig. 12a, b)

4e: ATGL immunoreactivity is indeed reduced in the LD and cytosolic fraction of *Grasp55*^{-/-} samples but unchanged in total membranes, which include the Golgi. Do the authors have an explanation for this discrepancy? Moreover, why is there β -actin detectable in the LD fraction?

>>Response:

1) As pointed out by the Reviewer, the total membrane (TM) fraction of ATGL showed no difference between the *Grasp55*^{+/+} and *Grasp55*^{-/-} samples (new Fig. 5d; previous Fig. 4e). We think that this is likely due to ATGL in the ER. As shown in Fig. 5a and Supplementary Fig. 9a, ATGL levels in the ER were not reduced upon GRASP55 deletion.

2) It has been reported that cytoskeletal proteins including beta-actin can be observed in the LD fraction using LC-MS/MS (PMID: 17004324; PMID: 17608402). However, in order to avoid confusion, we have replaced the beta-actin blot with that of Aldolase A, a cytosolic marker, which clearly shows absence of contamination with cytosolic proteins in the LD fraction (new Fig. 5d). Furthermore, we normalized ATGL levels in the LD using the LD marker RAB18 instead of beta-actin in the quantitative analyses; this more clearly shows the reduced ATGL content in *Grasp55*^{-/-} LDs (new Fig.5e).

To figure 5:

The immunostaining of ATGL in Caco-2 cells is of poor quality. ATGL immunoreactivity is detected all over the cell (even in the nucleus) but is hardly enriched specifically at the LDs. It is questionable, whether this staining permits a reliable readout of LD targeting. The authors need to use cell fractionation as a complementary method to validate this finding.

>>Response:

1) We have re-performed the immunostaining with a more optimized protocol, a higher concentration of primary antibodies (1:100) and a lower gain in confocal laser microscopy (Fig. 6c). We believe that the newer images showed an enrichment of ATGL at the LD periphery, in particular in control Caco-2 cells.

2) As requested, we have also performed the fractionation study in Caco-2 cells. The results clearly indicate that GRASP55 depletion results in reduced ATGL levels in LDs (Fig. 6a, b). Relevant comments have now been added in the text (lines 196-197).

5f,g: A reciprocal IP using anti-ATGL would substantiate the surprising and interesting finding that *Grasp55* and ATGL physically interact.

>>Response:

As suggested, we have now performed IP using antibodies against ATGL (anti-Myc with ATGL-Myc expression; Fig. 7c, d). The results further confirm that GRASP55 physically interacts with ATGL. Furthermore, a pull-down assay revealed that the GRASP55-PDZ1 domain specifically interacts with the ATGL-patatin domain (Fig. 7e, f). Relevant comments have now been added in the text (lines 209-218).

To line 194/195: What is the evidence from this work that specifically the “non-LD-associated ATGL” is subject to proteasomal degradation (and not also the LD pool)?

>>Response:

We stated this point based on the present finding that 1) GRASP55 deficiency reduced LD localization of ATGL (Fig. 6a–c), and 2) the reduced ATGL protein level was rescued by the proteasomal inhibitor MG132 (Supplementary Fig. 15). In addition, a previous study also suggested that “non-LD-associated ATGL is sensitive to proteasomal degradation” (Fig. 3 in Ref. #27). To avoid overstatement, we have now revised the text to indicate that parts of our results are consistent with a previous report, as follows: “These results are consistent with a previous report that non-LD-associated ATGL is sensitive to proteasomal degradation²⁷.” (lines 200-201).

To figure 6:

Ubiquitous over-expression of Grasp55 provides solid genetic evidence that the observed metabolic effects are indeed due to the knockout of Grasp55. Also, there is convincing evidence that lack of Grasp55-dependent lipid absorption, involving the proposed mechanism of lipase localization/protein turnover, contributes to the mutant phenotype. Yet in the absence of tissue-specific rescue experiments the impact of this mechanism on the mutant phenotype remains elusive.

>>Response:

We agree with the Reviewer that tissue-specific rescue experiments would further strengthen our conclusion. Instead of tissue-specific rescue experiments in mice, which would take a very long time, we performed some additional rescue experiments in Caco-2 cells (e.g., exogenous ATGL expression in GRASP55-depleted Caco-2 cells, Fig. 6f,g; see response to Minor Point 7 of Reviewer 2,) and also in flies (See response to Comments to Figure 7). All these results further support our conclusion that the loss of GRASP55 (or dGRASP) induces intestinal lipid absorption defects principally via defects in ATGL function. We are hopeful that

these additional results address, at least partially, the Reviewer's concerns regarding tissue-specific rescue experiments.

How does the protein expression level from the myc-tagged Grasp55 transgene relate to the endogenous expression? Figure 6h,i seems to indicate that intestinal ATGL abundance appears to be higher in the rescued mutants compared to controls.

>>Response:

1) See also response to Minor Point 6 of Reviewer 2. We have included immunoblot images showing the difference in protein levels between exogenous and endogenous GRASP55s in rescued mice (new Fig. 8h). Quantitative analysis indicated that the protein expression of exogenous GRASP55 was approximately eight times higher than that of endogenous GRASP55 in the mouse jejunum (lines 414-415).

2) As pointed out by the Reviewer, intestinal ATGL abundance appears to be higher in *TgGRASP55* mice. However, statistical analyses using the data shown in Fig. 8i revealed that the difference did not reach the statistical significance, under both control (lane 1 vs. lane 5, $P = 0.200$) and lipid-rich (lane 3 vs. lane 7, $P = 0.059$) conditions.

To figure 7:

The *Drosophila* data represent a meaningful contribution to the study. However, important details are not described and there is no evidence for an epistatic relation between *dGrasp* and *bmm* in the control of midgut lipid homeostasis. In particular, the following questions need to be addressed:

Where along the midgut does lipid accumulate in the *dGrasp* mutants? Do midgut enterocytes or other cell types accumulate lipids in this mutant condition?

>>Response:

The *Drosophila* midgut epithelium is composed of four cell types: intestinal stem cells, absorptive enterocytes, secretory enteroendocrine cells, and enteroblasts (PMID: 24016187). Under normal conditions, lipids are absorbed by enterocytes (PMID: 25263556). To resolve the Reviewer's question, we performed fly studies with enterocyte-specific GFP labeling and LD staining. The results indicated that *dGrasp* mutant flies accumulated lipids in the posterior midgut enterocytes after a standard diet feeding (new Fig. 9a, b). Relevant comments have now been added in the text (lines 262-264).

Is Bmm expressed in these cells? How does the bmm mutant phenotype look like in these cells/this region of the midgut? If bmm were downstream of dGrasp, (i) bmm mutants are expected to show a dGrasp-like ectopic lipid accumulation phenotype, and (ii) the double mutant could be reverted by providing bmm but not by providing dGrasp expression under the control of NP1.

>>Response:

We thank the Reviewer for this important suggestion. We have now performed a series of additional experiments to address the Reviewer's questions. It has been shown that BMM is expressed in the fly gut (PMID: 17488184). Notably, as shown in new Supplementary Fig. 21, *Bmm*^{-/-} flies, obtained from Dr. Ronald P. Kühnlein (Ref. # 17), showed a lipid accumulation phenotype in their posterior midguts, similar to *dGrasp*^{-/-} flies. The intestinal lipid accumulation phenotype in *Bmm*^{-/-} flies was reversed by the introduction of wild-type *Bmm* under the control of the enterocyte specific driver *NP1-GAL4*, further supporting the idea that BMM plays a role in lipid absorption in fly enterocytes. Interestingly, in contrast to BMM supplementation, the lipid accumulation phenotype of *Bmm*^{-/-} flies was not rescued by dGRASP supplementation (Supplementary Fig. 21). Considering the finding that the lipid accumulation phenotype of *dGrasp*^{-/-} flies was rescued by BMM supplementation (new Fig. 9c, d; previous Fig. 7a, b), these results imply that *Bmm* is epistatic to, and probably downstream of *dGrasp*. Relevant comments have now been added in the text (lines 281-283).

Rajan et al. (ref. 18; fig. 2B) report reduced body fat content in response to fat body (=adipose tissue+liver)-specific dGrasp55 in vivo knockdown, which ? at least for flies - challenges the model of the authors that adipose tissue hypotrophy in Grasp55 mutant mice is secondary to the lipid absorption defect (and not a consequence of cell autonomous effects in this tissue). These results also need to be discussed in the light of the reversion of the body fat content of dGrasp mutant flies subject to NP1-driven intestinal dGrasp expression.

>>Response:

As pointed out by the Reviewer, a recent study by Rajan *et al.* (Ref. # 18) reported that deletion of *dGrasp* in flies reduced body TG contents, similar to the results of the present study. However, in that study, the fat storage defect of *dGrasp*-deleted flies was rescued by fat body-specific dGRASP supplementation, in contrast to the

present finding that the fat storage defect of *dGrasp*-deleted flies was rescued by enterocyte-specific dGRASP supplementation (Fig. 9e). At present, the reason for the discrepancies between the two studies is unclear. A potential mechanism could be an age-dependent effect; Rajan *et al.* found that the *dGrasp* deletion effect on fat bodies was more profound with age. These points are now discussed in the text (lines 350-356).

General remarks:

Line 36: “gene silencing” is an inappropriate description for a deletion mutant.

>>Response:

Thank you. The term “silencing” has now been revised to “deletion” (line 33).

The authors are advised to carefully reinspect their claims concerning the cell/tissue-specificity of their experiments. E.g. they refer to enterocytes (line 179) but use complete intestine samples.

>>Response:

Where applicable throughout the manuscript, the term “enterocytes” has been revised to “intestine” or more specifically to “jejunum”.

Line 261: Beller et al. PLoS Biology 2008 already related ATGL degradation to failure in COPI trafficking.

>>Response:

In the previous manuscript, we had cited a review article from the same author group (Ref. #37). In this revised submission, we cited the original article describing COPI and ATGL function, rather than the review article, as suggested by the Reviewer (Reference #35, lines 302-303).

Attention needs: figure 5f ?IP:Ani-Myc“, “evolutionally” in line 231, “upstream activation signal” line 821.

>>Response:

Thank you so much for your careful review. The typographical errors have now been corrected to “Anti-Myc” (new Fig. 7a), “evolutionarily” (line 258), and “upstream activation sequence” (new Fig. 9 legend).

TG content measurements on flies appears to be missing from the Methods

section.

>>Response:

These are now described in the Methods section (lines 657-672).

The authors are advised to avoid calling w[1118] mutant flies wild-types. Designation as controls is correct and informative.

>>Response:

In the fly studies, the term “wild-type” has now been revised to “control” in the Results (line 264), Methods (lines 638-640), and Figures (Fig. 9).

The manuscript would benefit from another round of English proofreading to avoid e.g. ?... except for defected spermatogenesis ...“ (line 53).

>>Response:

As suggested, the manuscript has now been additionally proofread by a colleague who is an expert in scientific English. The sentence “except for defected spermatogenesis” has now been revised (line 61).

Reviewers' comments:

Reviewer #1 (Remarks to the Author):

The authors response to my concerns have been met.

Doug Mashek

Reviewer #2 (Remarks to the Author):

The authors have performed a significant number of experiments and revised a significant portion of the manuscript post-review. However, there is still a lack of mechanism, and several key points have not been addressed. Below are the major criticisms that relate to the original comments and rebuttal after reviewing the revised manuscript and figures:

1. There is still no mechanism, major criticism from the last review. The authors convincingly showed that the GRASP55 KO mice are lean, with the main cellular phenotypes including decreased LD number but increased size, and reduced ATGL and MGL protein levels and association with the Golgi, and decreased chylomicron secretion. But there is no actual mechanism provided on how GRASP55 deletion causes these phenotypes.
2. Related to Point 1 above for ATGL degradation, rather than speculating, the authors could test the effect of MG132 on the localization of ATGL to DLs, since they show that MG132 increases ATGL levels after GRASP55 shRNA (Figure S15). Particularly under the conditions of Figure S15 using knockdown and add-back of GRASP55 mutants. Furthermore, the authors don't show convincing evidence that ADARP levels and localization aren't changing in GRASP55^{-/-} mice, so using it as a colocalization marker is not ideal based on the images that are shown.
3. The authors cite Supplementary Fig. 19e,f and state: "Therefore, increased TG levels, rather than the GRASP55 expression status, appear to be primarily responsible for the increased LD size but reduced LD number in Grasp55^{-/-} enterocytes." To support their claim, the authors should have artificially manipulated the level of TG in GRASP55^{-/-} mice to test the size and number of LDs.
4. The conclusion that GRASP KO reduces ATGL and MGL association with LD in Figure 5 is questionable. Given that samples are loaded on top of the gradient, soluble proteins and low-density organelles such as LDs will both stay in the top fractions. Therefore the statement that there is less

ATGL and MGL associated with LD is invalid, in particular, that the total amount of these proteins is significantly reduced. For a purpose like this, floatation gradients are more often used to isolate LDs. In the fluorescence images, given that ATGL is distributed in the entire cytoplasm as shown in this study, it is hard to conclude that the reduces fluorescence signal that overlaps with ADRP (Figure 5f) or BODIPY dye (Figure 6c) is due to the reduced association or simply because the total protein level is lower in the cell.

5. As pointed out in the last review, GRASP55 is involved in autophagy and Figure 3g shows some membrane profiles like autophagosomes. The authors provided a new blot on LC3 in Figure S18 in which LC3 II/I ratio does not change. This blot is overexposed and a minor, but clear difference in the ratio of LC3-1/LC3-11 is evident. In this blot, the level of LC3-II is higher than LC3-I, different from many publications of LC3 blot from tissues (such as Benjamin et al., Cell Host Microbe. 2013) showing that over 90% of LC3 in tissues including intestine epithelial cells exist as LC3-I.

6. Figure S3 (as well as Figure 3g) shows an EM image in which the Golgi stack is abnormal, but this abnormality is not quantified nor discussed. The alignment of the cisternae is different from that in the control mice. If this is not considered as a stacking defect, then what is it? What does it mean that the luminal width is increased in related to the Golgi function? Also, it is not clear how representative this image is; displaying a gallery of EM images is needed.

7. Some of the quantitation results do not match the western blots. For example, Figure S15, although the ARGL blot seems overexposed, the level of ATGL in the “-” lanes appears to be lower than the next “+” lane; but this is not reflected in the quantitation. Another example is Figure 6d,e, the reduction rate in all 4 sets seems the same on the gel but the quantitation shows a significant change. Similar concerns exist on Figure 8 h,i.

8. The different effects of GRASP55 KD in LD biogenesis in different tissues increase the difficulty to understand the exact role of GRASP55 in this process. As Reviewer 3 requested, a further study on different tissues is required for understanding the role of GRASP55 in lipid metabolism.

9. Some of the supplemental figures can be combined when they are showing related results.

Reviewer #3 (Remarks to the Author):

The revised version of the manuscript by Kim and colleagues on Grasp55 function in the context of fat metabolism satisfyingly addresses all important issues raised by this reviewer.

Congratulations to the authors for an (almost too) extensive, very interesting study.

A final detail for improvement. The authoritative *Drosophila* database FlyBase lists the fly ATGL homolog brummer (bmm) as recessive gene. Accordingly, the gene name and the gene name abbreviation (in contrast to the corresponding proteins) start with lowercase letter.

Response to Reviewers' Comments

Reviewer #1

The authors response to my concerns have been met.

>> **Response:** We wish to thank Reviewer 1 for taking the time to thoroughly review our manuscript and for previous insightful comments, which substantially improved the overall quality of this study.

Reviewer 2

1. There is still no mechanism, major criticism from the last review. The authors convincingly showed that the GRASP55 KO mice are lean, with the main cellular phenotypes including decreased LD number but increased size, and reduced ATGL and MGL protein levels and association with the Golgi, and decreased chylomicron secretion. But there is no actual mechanism provided on how GRASP55 deletion causes these phenotypes.

>> **Response:** We would like to thank Reviewer 2 for taking the time to thoroughly review our manuscript and for valuable comments. We have performed additional experiments to elucidate the mechanism by which GRASP55 regulates intestinal lipid absorption at the molecular level.

1) First, we examined the effects of proteasomal inhibitor MG132 on ATGL levels in the cytosol and LDs (Fig. 8a–d). Depletion of GRASP55 reduced ATGL levels in both the cytosol and LDs. Notably, MG132 reversed the decreased cytosolic levels of ATGL (Fig. 8a, b), but did not affect ATGL levels in LDs of the GRASP55-depleted cells (Fig. 8c, d). In contrast, the add-back of GRASP55 restored ATGL levels in both the cytosol and LDs (Fig. 8a–d). These results

indicate that the simple restoration of cytosolic ATGL to levels near control levels is not sufficient to overcome LD localization defects of ATGL caused by GRASP55 depletion.

2) We then analyzed the association between GRASP55 and ATGL after blockade of ER-to-Golgi trafficking by dominant-inhibitory forms of SAR1 (SAR1-T39N) and ARF1 (ARF1-T31N). Notably, the interaction between GRASP55 and ATGL was abolished by the SAR1 and ARF1 mutants, suggesting that transport of ATGL to the Golgi is required for association between ATGL and GRASP55 (Fig. 8e, f).

3) The ER-to-Golgi blockade by SAR1 and ARF1 inhibition also reduced ATGL levels in the cytosol and LDs (Supplementary Fig. 14). MG132 did not recover the reduced ATGL levels in LD, but did increase cytosolic ATGL levels in these cells (Supplementary Fig. 14). This result is identical to the result obtained in GRASP55-depleted cells (Fig. 8a–d), again indicating that ATGL levels in LDs do not simply reflect ATGL levels in cytosol.

4) The SAR1 and ARF1 inhibition reduced LD localization of RAB18 and ADRP as well as that of ATGL (Supplementary Fig. 14), suggesting that global blockade of ER-to-Golgi transport inhibits LD localization of many LD-associated proteins. In contrast, GRASP55 depletion inhibited LD localization only of ATGL without affecting that of RAB18 or ADRP, indicating that GRASP55 specifically regulates LD localization of ATGL (Supplementary Fig. 14).

In aggregate, these results imply that defective LD localization of ATGL is the primary event caused by GRASP55 depletion, and that the restoration of cytosolic ATGL levels to near control levels is not sufficient to recover the LD trafficking defect of ATGL. Decreased ATGL levels in LDs will eventually cause the formation of supersized LDs and decreased chylomicron secretion due to reduction in lipid mobilization of LDs.

We have added descriptions and discussion of these results to the Results section (lines 216–238) and Discussion section (lines 341–347), respectively.

2. Related to Point 1 above for ATGL degradation, rather than speculating, the authors could test the effect of MG132 on the localization of ATGL to DLs, since they show that MG132 increases ATGL levels after GRASP55 shRNA (Figure S15). Particularly under the conditions of Figure S15 using knockdown and add-back of GRASP55 mutants. Furthermore, the authors don't show convincing evidence that ADARP levels and localization aren't changing in GRASP55^{-/-} mice, so using it as a colocalization marker is not ideal based on the images that are shown.

>> **Response:** As described in the Response to Question 1, we have examined the effects of MG132 on LD targeting of ATGL in Caco-2 cells. While MG132 reversed the decreased cytosolic levels of ATGL (Fig. 8a, b; corresponding to previous Figure S15), it did not affect ATGL levels in LDs in the GRASP55-depleted cells (Fig. 8c, d).

We have re-blotted the WB membranes used in Fig. 5a, b to detect ADARP levels. The results, presented in new Fig. 5a, indicated that fraction 1 contains the LD marker protein ADARP. Furthermore, the results in new Fig. 5b and Supplementary Fig. 14a showed that LD localization and protein levels of ADARP did not change in GRASP55 KO mice and GRASP55-depleted Caco-2 cells, respectively. Relevant descriptions were added in the main text (lines 240 and 341-345) and the legend for Fig. 5 and Supplementary Fig. 14.

3. The authors cite Supplementary Fig. 19e,f and state: "Therefore, increased TG levels, rather than the GRASP55 expression status, appear to be primarily responsible for the increased LD size but reduced LD number in Grasp55^{-/-} enterocytes." To support their claim, the authors should have artificially manipulated the level of TG in GRASP55^{-/-} mice to test the size and number of LDs.

>> **Response:** We agree with the Reviewer that a more sophisticated biophysical experiment is required to fully resolve this question. However, we believe that there is a miscommunication between the Reviewer and us. Our statement of "Therefore, increased TG levels..." was presented in our previous Response to Reviewers' Comments, not in the manuscript. For the

purpose of answering Reviewer 2's earlier question of "Similarly, how does GRASP55 increases the size but reduces the number of LD?", we simply accepted Reviewer 2's implication that GRASP55-deleted enterocytes have a reduced number of LDs on exogenous lipid challenge. However, we do not think that the present study substantiates that implication.

Please note that we analyzed the frequency distribution of LD size in a given number of LDs in the EM images (Fig. 3f), but did not analyze the total number of LDs in a cell. In a preliminary analysis, we found that the cellular LD numbers were too variable to draw a conclusion, although the cells with supersized LDs appeared to have a decreased number of LDs. Therefore, in our main text and Supplementary Information, we described that "*Grasp55*^{+/+} mouse enterocytes showed supersized LDs and increased TG levels on exogenous lipid challenges," without commenting on cellular LD numbers.

In the new Supplementary Fig. 17f (previous Supplementary Fig. 19), we have provided the results of LD diameter frequency analyses in hepatocytes. In hepatocytes, increased TG levels, but not GRASP55 deficiency, were associated with an increase in LD size (Supplementary Fig. 17e-g). Therefore, both sets of results (in *Grasp55*^{+/+} hepatocytes and in *Grasp55*^{-/-} enterocytes) were consistent: increased TG levels, rather than GRASP55 expression status, were associated with increased LD size.

4. The conclusion that GRASP KO reduces ATGL and MGL association with LD in Figure 5 is questionable. Given that samples are loaded on top of the gradient, soluble proteins and low-density organelles such as LDs will both stay in the top fractions. Therefore the statement that there is less ATGL and MGL associated with LD is invalid, in particular, that the total amount of these proteins is significantly reduced. For a purpose like this, floatation gradients are more often used to isolate LDs. In the fluorescence images, given that ATGL is distributed in the entire cytoplasm as shown in this study, it is hard to conclude that the reduces fluorescence signal that overlaps with ADRP (Figure 5f) or BODIPY dye (Figure 6c) is due to the reduced association or

simply because the total protein level is lower in the cell.

>> **Response:** We thank the Reviewer for these careful comments.

1) For Fig. 5, we prepared the LD fractions using the flotation gradients protocol, which we have now fully described in the Methods section (lines 505–508). In mouse epithelia, we first isolated LDs by obtaining the top of flotation gradients (Ref. #47). We then subjected the remaining parts to subcellular fractionation using the OptiPrep density gradient. Therefore, the fractions labeled “Fraction 1” in Fig. 5a in fact represent proteins in LDs, and were not derived from the continuous density gradients. This distinction is now indicated in the legend for Fig. 5a.

2) We agree with the Reviewer that reduced total ATGL levels may also contribute to the reduced fluorescence signal that overlaps with LD markers. To address this question, we provided the results of additional MCC analysis measuring “ATGL colocalized with ADRP/total ATGL”, which should be independent of total ATGL levels (Fig. 5g, right). Again, the results indicated that the association between ATGL and ADRP is reduced in *Grasp55^{+/+}* mouse enterocytes regardless of the ATGL levels.

5. As pointed out in the last review, GRASP55 is involved in autophagy and Figure 3g shows some membrane profiles like autophagosomes. The authors provided a new blot on LC3 in Figure S18 in which LC3 II/I ratio does not change. This blot is overexposed and a minor, but clear difference in the ratio of LC3-1/LC3-11 is evident. In this blot, the level of LC3-II is higher than LC3-I, different from many publications of LC3 blot from tissues (such as Benjamin et al., Cell Host Microbe. 2013) showing that over 90% of LC3 in tissues including intestine epithelial cells exist as LC3-I.

>> **Response:** We thank the Reviewer for this insightful question. In our previous experiments, we prepared control mouse samples after overnight starvation. This appeared to be responsible for higher LC3-I levels. In the new Supplementary Fig. 16, we have provided the results of LC3 analyses using samples from ad libitum-fed animals. The results indicated that *Grasp55* deletion

did not significantly affect autophagic flux in the mouse jejunum.

6. Figure S3 (as well as Figure 3g) shows an EM image in which the Golgi stack is abnormal, but this abnormality is not quantified nor discussed. The alignment of the cisternae is different from that in the control mice. If this is not considered as a stacking defect, then what is it? What does it mean that the luminal width is increased in related to the Golgi function? Also, it is not clear how representative this image is; displaying a gallery of EM images is needed.

>> **Response:** We think the Reviewer is referring to Supplementary Fig. 7 in the previous manuscript. As requested, we have now provided a gallery of EM images and a quantification of the maximum luminal width of the Golgi stacks in the new Supplementary Fig. 6. The GRASP55 deficiency in mouse enterocytes appeared to increase the maximum luminal width of the Golgi stacks (from 49.9 ± 2.2 nm to 58.1 ± 3.8 nm, $p = 0.065$) without causing Golgi unstacking. These results are consistent with a previous study showing that single knockdown of GRASP55 among the four Golgi stacking factors (GM130, Golgin45, GRASP65, and GRASP55) results in a 17% increase in the maximum luminal width without significant Golgi unstacking in HeLa cells (Ref. #21, Fig. 1). In order to precisely represent our findings, we have revised our previous description (“Although some Golgi membranes appeared to have lost their characteristic flattened cisternae, the depletion of GRASP55 alone did not disrupt the Golgi stacking structures”) to “Although some Golgi membranes appeared to have lost their characteristic flattened cisternae, the GRASP55 deficiency alone did not cause Golgi unstacking” (lines 165–167).

The functional relevance of the Golgi flatness defects to chylomicron secretion is now discussed in the text (lines 347–351). Although we cannot completely rule out the possibility that Golgi flatness defects contribute to the reduced chylomicron secretion, such contribution appeared to be minimal because only 7.5% of the Golgi stacks showed an abnormal enlargement (>0.1 μm).

7. Some of the quantitation results do not match the western blots. For example, Figure S15, although the ARGL blot seems overexposed, the level of ATGL in the "-" lanes appears to be lower than the next "+" lane; but this is not reflected in the quantitation. Another example is Figure 6d,e, the reduction rate in all 4 sets seems the same on the gel but the quantitation shows a significant change. Similar concerns exist on Figure 8 h,i.

>> **Response:** We have replaced the immunoblot images of Figs. 6d, 8a (previous Supplementary Fig. 15a), and 9h (previous Fig. 8h) with less exposed images. In addition, we have re-performed the quantitation of Fig. 6d with more careful inspection of the automated region of interest (ROI) selection and the results are presented in new Figs. 6e.

8. The different effects of GRASP55 KD in LD biogenesis in different tissues increase the difficulty to understand the exact role of GRASP55 in this process. As Reviewer 3 requested, a further study on different tissues is required for understanding the role of GRASP55 in lipid metabolism.

>> **Response:** In our previous Response to Reviewers' Comments, we addressed this question raised by Reviewer 3 and added relevant discussion to the text (lines 379-385). A recent study by Rajan et al. (Ref. #18) reported that deletion of *dGrasp* in flies reduced body TG contents, similar to the fly experiment results of the present study. However, in the Rajan study, the fat storage defect of *dGrasp*-deleted flies was rescued by fat body-specific *dGrasp* supplementation, in contrast to the present finding that the fat storage defect of *dGrasp*-deleted flies was rescued by enterocyte-specific *dGrasp* supplementation (Fig. 10e). Although factors such as an age-dependent effect may explain the differences, the reason for the discrepancies between the two studies is unclear, because we do not know the exact experimental conditions of the study by Rajan et al. However, we believe that our manuscript has its own merits, because it provides integrated information through analyses in mouse, fly, and *in vitro* systems. In all experiments,

the results indicate that GRASP55 plays an important role in intestinal fat absorption by facilitating LD localization of some LD proteins, such as ATGL and MGL. We hope that these results will contribute to understanding the role of GRASP55 and the Golgi in lipid metabolism.

9. Some of the supplemental figures can be combined when they are showing related results.

>> **Response:** As suggested, we have now combined several Supplementary Figures that show related results. Data from previous Supplementary Figs. 4 and 5 have been combined into a new Supplementary Fig. 4, and data from Supplementary Figs. 12 and 13 have been combined into a new Supplementary Fig. 11.

Reviewer #3

The revised version of the manuscript by Kim and colleagues on Grasp55 function in the context of fat metabolism satisfyingly addresses all important issues raised by this reviewer.

Congratulations to the authors for an (almost too) extensive, very interesting study.

A final detail for improvement. The authoritative Drosophila database FlyBase lists the fly ATGL homolog brummer (*bmm*) as recessive gene. Accordingly, the gene name and the gene name abbreviation (in contrast to the corresponding proteins) start with lowercase letter.

>> **Response:** We thank Reviewer 3 for taking the time to thoroughly review our manuscript and for insightful comments, which significantly improved the overall quality and accuracy of this study. As recommended, we revised the gene name and its abbreviation '*bmm*' starting with lowercase letter in entire text and figures.

REVIEWERS' COMMENTS:

Reviewer #2 (Remarks to the Author):

The authors have performed a number of new experiments to address my comments. I have no new questions.

Some of the figures/images have low resolutions and should be improved.

Response to Reviewer's Comments

Reviewer #2 (Remarks to the Author):

The authors have performed a number of new experiments to address my comments. I have no new questions.

Some of the figures/images have low resolutions and should be improved.

>> **Response:** We are pleased that Reviewer 2 is satisfied with the revisions to the manuscript. We previously made figures in a 300-dpi format according to the Journal guidelines. Now, we provide all main and supplementary figures in a 600-dpi format to improve the resolution of figures and images.